# The burden of post-acute COVID-19 symptoms in a multinational network cohort analysis

Kristin Kostka[1,10], Elena Roel[2,3,10], Nhung T. H. Trinh [4], Núria Mercadé-Besora[2], Antonella Delmestri [1], Lourdes Mateu[5,6], Roger Paredes [5,6,7,8], Talita Duarte-Salles[2,9], Daniel Prieto-Alhambra [1,9] ✉, Martí Català[1,11] & Annika M. Jödicke[1,11]

Persistent symptoms following the acute phase of COVID-19 present a major burden to both the affected and the wider community. We conducted a cohort study including over 856,840 first COVID-19 cases, 72,422 re-infections and more than 3.1 million first negative-test controls from primary care electronic health records from Spain and the UK (Sept 2020 to Jan 2022 (UK)/March 2022 (Spain)). We characterised post-acute COVID-19 symptoms and identified key symptoms associated with persistent disease. We estimated incidence rates of persisting symptoms in the general population and among COVID-19 patients over time. Subsequently, we investigated which WHO-listed symptoms were particularly differential by comparing their frequency in COVID-19 cases vs. matched test-negative controls. Lastly, we compared persistent symptoms after first infections vs. reinfections. Our study shows that the proportion of COVID-19 cases affected by persistent post-acute COVID-19 symptoms declined over the study period. Risk for altered smell/taste was consistently higher in patients with COVID-19 vs test-negative controls. Persistent symptoms were more common after reinfection than following a first infection. More research is needed into the definition of long COVID, and the effect of interventions to minimise the risk and impact of persistent symptoms.

Three years after the world's first reported cases of the novel coronavirus (SARS-CoV-2 and ensuing global pandemic), COVID-19 is still a significant burden to morbidity and mortality globally[1]. As early as May 2020[2], clinicians observed a subset of COVID-19 cases that evolved from an acute viral infection into a long-term condition with a puzzling array of symptoms, subsequently called "long COVID"[3]. Many have sought to standardise how clinicians distinguish post-acute COVID-19 sequelae from initial infection and ongoing symptomatic COVID-19[4–6]. Through a Delphi consensus process[7], the World Health Organisation (WHO) defined 'post COVID-19 condition' as a condition that occurs in individuals with a probable or confirmed history of SARS-CoV-2 infection who present at least 3 months from the onset of COVID-19 with new or persisting symptoms for at least 2 months that cannot be attributed to another aetiology. Common persisting

[1]Pharmaco- and Device Epidemiology Group, CSM, NDORMS, University of Oxford, Oxford, United Kingdom. [2]I Fundació Institut Universitari per a la recerca a l'Atenció Primària de Salut Jordi Gol i Gurina (IDIAPJGol), Barcelona, Spain. [3]Universitat Autònoma de Barcelona, Bellaterra (Cerdanyola del Vallès), Barcelona, Spain. [4]PharmacoEpidemiology and Drug Safety Research Group, Department of Pharmacy, University of Oslo, Oslo, Norway. [5]Department of Infectious Diseases, Hospital Germans Trias i Pujol, Badalona, Spain. [6]Fundació Lluita contra les Infeccions, Badalona, Spain. [7]irsiCaixa AIDS Research Institute, Badalona, Spain. [8]Center for Global Health and Diseases, Department of Pathology, Case Western Reserve University School of Medicine, Cleveland, OH, USA. [9]Department of Medical Informatics, Erasmus University Medical Center, Rotterdam, The Netherlands. [10]These authors contributed equally: Kristin Kostka, Elena Roel. [11]These authors jointly supervised this work: Martí Català, Annika M. Jödicke. ✉e-mail: daniel.prietoalhambra@ndorms.ox.ac.uk

symptoms include debilitating fatigue, shortness of breath, memory or cognitive dysfunction, and a variety of other multi-system symptoms[8] that affect day-to-day living. Symptoms can be new onset after initial recovery from the first SARS-CoV-2 infection or can persist from the infection and may fluctuate or relapse over time. Recent literature suggests significant heterogeneity in how individuals experience symptoms, including the emergence of potential clinical subgroups[9]. Therefore, quantifying post-acute COVID-19 condition is challenging: some reported symptoms are non-specific and prevalent in the general population regardless of infection status. A recent meta-analysis estimated that 6.2% of people with symptomatic SARS-CoV-2 infection self-reported post-acute COVID-19 symptoms, such as persistent fatigue with pain or mood swings, cognitive problems or ongoing respiratory symptoms, in 2020/2021[10].

Various single-country analyses have evaluated patterns of post-acute COVID-19 symptoms in specific care environments, each contributing new understanding to the incompletely understood natural history of post-acute COVID-19 sequalae[11,12]. However, only a few attempted to compare clinical definitions across multiple countries[13] and/or sources of real-world data.

For this study, we took advantage of large primary care electronic health records datasets from two European countries, namely CPRD AURUM (England, UK) and SIDIAP (Catalonia, Spain), to characterise post COVID-19 conditions and identify key symptoms associated with persistent disease. We first estimated age- and sex-specific incidence rates of persisting symptoms in the general population and among people with confirmed COVID-19 over time. Subsequently, we

investigated which of the 25 symptoms the WHO mentions in their clinical case definition are particularly specific to long COVID by comparing the occurrence of each symptom among COVID-19 patients and people who tested negative in the same week. Lastly, we compared the occurrence of persistent symptoms after a first infection or after reinfection with SARS-CoV-2.

## Results

### Characterisation of people with COVID-19 and negative-test comparator cohorts

We included 448,361 and 480,901 SARS-CoV-2 infections and 1,644,166 and 1,508,585 first SARS-CoV-2 negative test controls recorded during the study period in SIDIAP and CPRD AURUM, respectively. The study inclusion process is illustrated in Fig. 1. Baseline characteristics of both the SARS-CoV-2 infection cohort and the first SARS-CoV-2 negative test cohort are reported in Table 1. Overall, follow-up was similar in SIDIAP and CPRD, with a median of 1 year (median 342 days and 364 days for SARS-CoV-2 infection cohorts in SIDIAP and CPRD, respectively, and 365 days for first negative test controls for both databases). Participants from both databases were mostly young adults (<50 years), female and predominantly vaccinated. Among the latter, the proportion of just one dose of vaccination was higher in the SARS-CoV-2 infected compared to negative test controls, with higher proportions of booster doses in the negative test controls. Baseline characteristics for cohorts of first SARS-CoV-2 infections, SARS-CoV-2 re-infections, and all SARS-CoV-2 negative tests are provided in Supplementary Table S1.

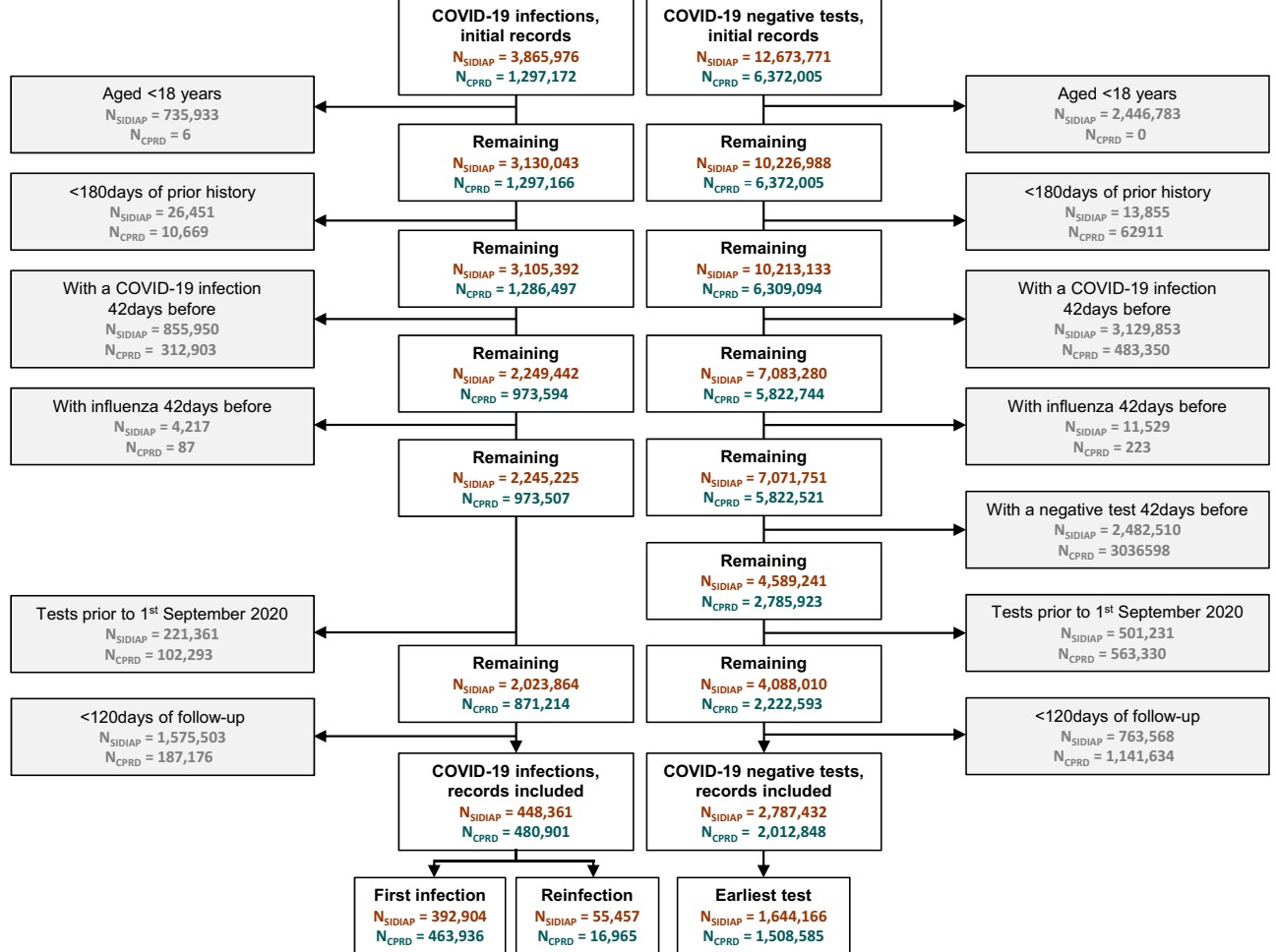

**Fig. 1 | Selection of participants from the SIDIAP and CPRD databases.** SIDIAP Sistema d'Informació per al Desenvolupament de la Investigació en Atenció Primària, CPRD Clinical Practice Research Datalink.

**Table 1 | Baseline characteristics in the SARS-CoV-2 infection and first SARS-CoV-2 negative test cohorts, by database**

| | SIDIAP | | CPRD | |
|---|---|---|---|---|
| | SARS-CoV-2 infection | First SARS-CoV-2 negative test | SARS-CoV-2 infection | First SARS-CoV-2 negative test |
| N | 448,361 | 1,644,166 | 480,901 | 1,508,585 |
| SARS-CoV-2 PCR test (%) | 233,281 (52.0) | 1,644,166 (100.0) | 338,504 (70.4) | 1,508,585 (100.0) |
| *Test date period (%)* | | | | |
| Sep–Dec 2020 | 127,419 (28.4) | 700,640 (42.6) | 211,657 (44.0) | 914,008 (60.6) |
| Jan–Apr 2021 | 137,539 (30.7) | 527,588 (32.1) | 186,027 (38.7) | 571,401 (37.9) |
| May–Aug 2021 | 159,904 (35.7) | 289,618 (17.6) | 74,405 (15.5) | 23,040 (1.5) |
| Sep–Dec 2021 | 23,499 (5.2) | 126,320 (7.7) | 8812 (1.8) | 136 (0.0) |
| *Wave (%)* | | | | |
| Wild | 106,231 (23.7) | 601,101 (36.6) | 148,901 (31.0) | 734,616 (48.7) |
| Alpha | 165,159 (36.8) | 667,662 (40.6) | 252,093 (52.4) | 768,109 (50.9) |
| Delta | 176,971 (39.5) | 375,403 (22.8) | 79,907 (16.6) | 5860 (0.4) |
| Days of follow-up (median [IQR]) | 342 [249,365] | 365 [276, 365] | 364 [300, 365] | 365 [334, 365] |
| Age (median [IQR]) | 42 [29, 56] | 47 [34, 62] | 41 [29, 54] | 41 [30, 55] |
| *Age, categories (%)* | | | | |
| ≤34 | 159,888 (35.7) | 423,729 (25.8) | 176,631 (36.7) | 529,987 (35.1) |
| 35–49 | 132,947 (29.7) | 472,047 (28.7) | 139,975 (29.1) | 457,006 (30.3) |
| 50–64 | 90,924 (20.3) | 390,178 (23.7) | 112,104 (23.3) | 360,586 (23.9) |
| 65–79 | 41,208 (9.2) | 258,997 (15.8) | 37,204 (7.7) | 127,153 (8.4) |
| ≥80 | 23,394 (5.2) | 99,215 (6.0) | 14,987 (3.1) | 33,853 (2.2) |
| Sex, male (%) | 195,901 (43.7) | 771,431 (46.9) | 218,169 (45.4) | 662,853 (43.9) |
| *Vaccination status (%)* | | | | |
| Not vaccinated | 80,053 (17.9) | 221,018 (13.4) | 75,583 (15.7) | 165,818 (11.0) |
| One dose | 149,014 (33.2) | 242,323 (14.7) | 71,496 (14.9) | 110,386 (7.3) |
| Two doses | 132,782 (29.6) | 475,664 (28.9) | 143,346 (29.8) | 475,537 (31.5) |
| Three or more (booster) doses | 86,512 (19.3) | 705,161 (42.9) | 190,476 (39.6) | 756,844 (50.2) |
| *Comorbidities (%)* | | | | |
| Asthma | 34,729 (7.7) | 124,092 (7.5) | 80,565 (16.8) | 253,906 (16.8) |
| Autoimmune disease | 8426 (1.9) | 35,035 (2.1) | 12,776 (2.7) | 40,531 (2.7) |
| COPD | 11,860 (2.6) | 61,399 (3.7) | 9463 (2.0) | 26,131 (1.7) |
| Dementia | 8377 (1.9) | 17,073 (1.0) | 5292 (1.1) | 13,666 (0.9) |
| Diabetes | 36,609 (8.2) | 156,220 (9.5) | 35,636 (7.4) | 95,640 (6.3) |
| Heart disease | 49,726 (11.1) | 219,520 (13.4) | 34,929 (7.3) | 99,275 (6.6) |
| Cancer | 29,106 (6.5) | 142,116 (8.6) | 21,259 (4.4) | 69,447 (4.6) |
| Hypertension | 76,227 (17.0) | 360,097 (21.9) | 66,130 (13.8) | 194,132 (12.9) |
| Renal impairment | 18,779 (4.2) | 83,165 (5.1) | 20,135 (4.2) | 51,474 (3.4) |

*SIDIAP* Sistema d'Informació per al Desenvolupament de la Investigació en Atenció Primària, *CPRD* Clinical Practice Research Datalink, *IQR* interquartile range, with q25 and q75 provided.

Post-acute COVID-19 symptoms and the distribution of ongoing symptoms across cohorts are reported in Supplementary Tables S2 and S3. Among those with a SARS-CoV-2 infection, 22.5% in SIDIAP and 21.0% in CPRD had post-acute COVID-19 symptoms, as defined by the presence of at least one of the 25 WHO-listed symptoms recorded ≥90 days after infection (and without a history of the symptoms in the 180 days before SARS-CoV-2 infection). In comparison, 21.3% of participants in the first negative test cohort in SIDIAP and 23.0% in CPRD had at least one persistent symptom recorded ≥90 days after the test (despite no infection). Across all cohorts and databases, more than ~60% of individuals with post-acute COVID-19 symptoms had only one symptom recorded. Overall, the most common symptoms were joint pain, abdominal pain, gastrointestinal issues, and anxiety. Cough and depression were common in CPRD. Sensitivity analyses showed that "ongoing symptoms" ≥28 days were more frequent than symptoms recorded at ≥90 days, with 28.0% of SARS-CoV-2 infections followed by at least one symptom recorded ≥28 days after SARS-CoV-2 infection in SIDIAP and 25.2% in CPRD.

## Incidence of post-acute COVID-19 symptoms across the two databases

Monthly incidence rates per 100,000 person-years of SARS-CoV-2 infection and post-acute COVID-19 symptoms ≥90 days after COVID-19 are shown stratified by age and sex in Fig. 2 and are shown overall in Fig. S2 and Table S4. The incidence rate of COVID-19 in each database mirrored the official rate of SARS-CoV-2 infections in that country. The incidence rate of post-acute COVID-19 in the general population mirrored and followed waves of COVID-19. However, rates of post-acute symptoms among individuals with COVID-19 declined over time in both databases. The rates of post-acute COVID-19 symptoms were clearly higher in younger people aged 18–34 and 35–49 years old, particularly during the summer of 2021.

## Risks of post-acute COVID-19 symptoms following COVID-19

We matched 1:3 by age group, sex, type of test (antigen or PCR) and index week 229,086 and 332,276 COVID-19 patients to 591,145 and 912,745 first negative test controls in SIDIAP and CPRD, respectively. Baseline characteristics were balanced for the matched cohorts, with

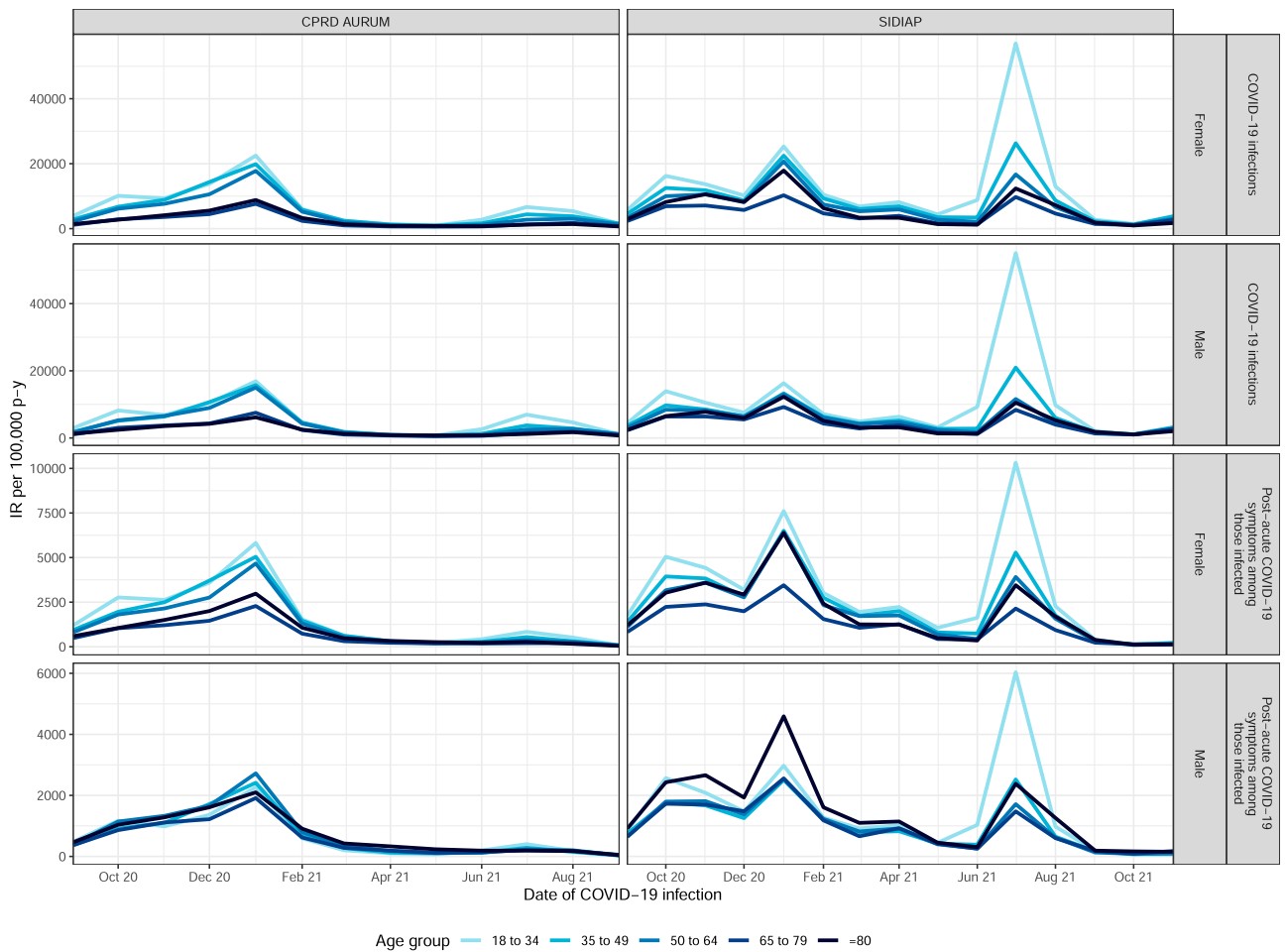

**Fig. 2 | Incidence rates of COVID-19 and post-acute COVID-19 symptoms ≥90 days after infection in the general population and among infected people over time, stratified by sex and age group.** IR incidence rate, p-y person-years.

standardised mean differences (SMD) < 0.1 for all covariates, except for days of follow-up (SMD 0.113) and the number of vaccine doses in SIDIAP (Supplementary Table S5). Figure 3 and Table S6 show rate ratios for each pre-defined symptom, comparing COVID-19 to matched test-negative cohorts. We found increased risks of persistent (≥90 days) altered smell and taste in both databases, with a rate ratio of 4.91 (95% CI: 4.08–5.90) in SIDIAP and 2.67 (2.35–3.03) in CPRD. Increased risk of fatigue/malaise was seen in CPRD (RR 1.06 [1.02–1.09]), and increased risks of dyspnoea (RR 1.12, [1.05–1.20]) was seen in SIDIAP. In CPRD, dyspnoea was more common in the negative-test cohort than among COVID-19 matched cases, inconsistent with findings from SIDIAP.

Results from stratification for a wave of predominant variant are in line with the overall findings and included in Fig. 4.

Results from sensitivity analyses for symptoms ≥28 days after SARS-CoV-2 infection or negative test are provided in Supplementary Fig. S3 and Table S7. Both databases showed consistent increased risks for altered smell/taste and fatigue. In SIDIAP, in addition to dyspnoea, increased risk after ≥90 days was also seen for menstrual problems and cough. Results from sensitivity analyses matching *any* negative test instead of the *first* negative test are included in Supplementary Tables S9 and S10 and Figs. S4 and S5 and showed similar findings.

### Post-acute COVID-19 symptoms associated with SARS-CoV-2 re-infection

We matched 155,400 and 48,574 first SARS-CoV-2 infections up to 3:1 to 55,297 and 16,916 reinfections in SIDIAP and CPRD,

respectively. Baseline characteristics were broadly balanced after matching, with SMD < 0.1 for demographics and all co-variates except for days of follow-up and number of vaccine doses in SIDIAP (SMD: 0.102 and 0.128, respectively) (Table S8). Figure 5 illustrates that the risk of post-acute COVID-19 symptoms was consistently increased after re-infection, compared to after the first infection (Table S11). Results from sensitivity analyses with symptoms assessed after ≥28 days showed a similar trend (Fig. S6, Table S12).

## Discussion

### Statement of principal findings

This multinational cohort study characterises the presentation of long COVID, including over 856,840 first COVID-19 cases, 72,422 re-infections and more than 3.1 million first negative-test controls from Spain and the UK. We found high proportions of post-acute COVID-19 symptoms for ≥90 days, i.e. after almost 22.5% of infections in Spain and 21% in the UK. At the population level, waves of post-acute COVID-19 symptoms followed each wave of community transmission in the study period, affecting predominantly young adults. However, the proportion of people infected with COVID-19 who went on to develop post-acute COVID-19 symptoms declined over time.

At the patient level, some persisting symptoms appeared more specific and differential for post-COVID-19 infection when compared to matched contemporaneous negative-test controls. Altered smell and taste was consistently more common

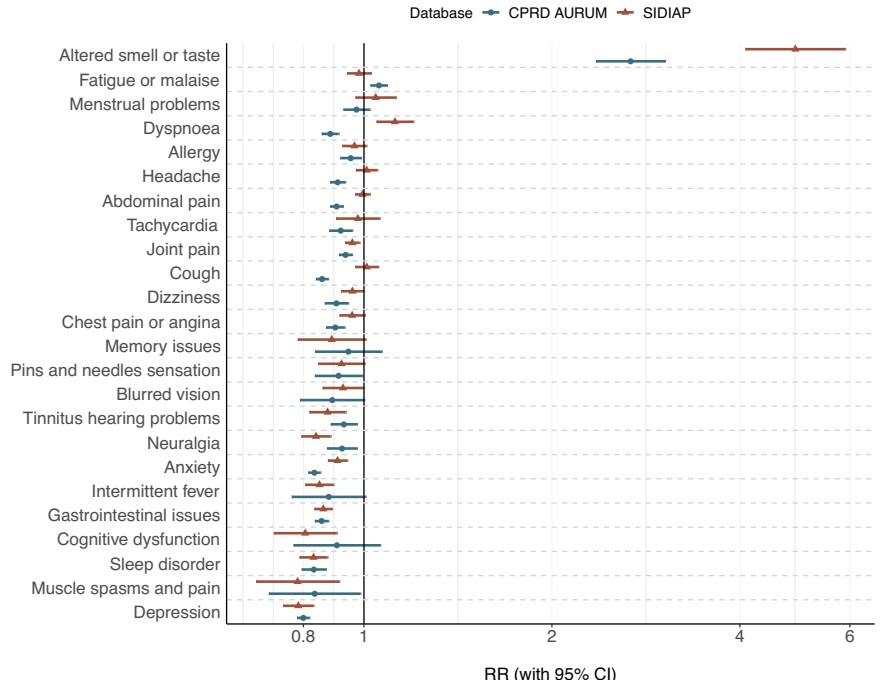

**Fig. 3 | Rate ratios (RRs) for pre-defined post-acute COVID-19 symptoms, comparing 1:3 matched SARS-CoV-2 infections to first SARS-CoV-2 negative tests.** SIDIAP Sistema d'Informació per al Desenvolupament de la Investigació en Atenció Primària, CPRD Clinical Practice Research Datalink, RR Rate ratio, 95% CI 95% confidence intervals. RR and 95% CI were calculated among 229,086 COVID-19 infections matched to 591,145 first SARS-CoV-2 negative tests in SIDIAP and 332,276 COVID-19 infections matched to 912,745 first SARS-CoV-2 negative tests in CPRD, respectively.

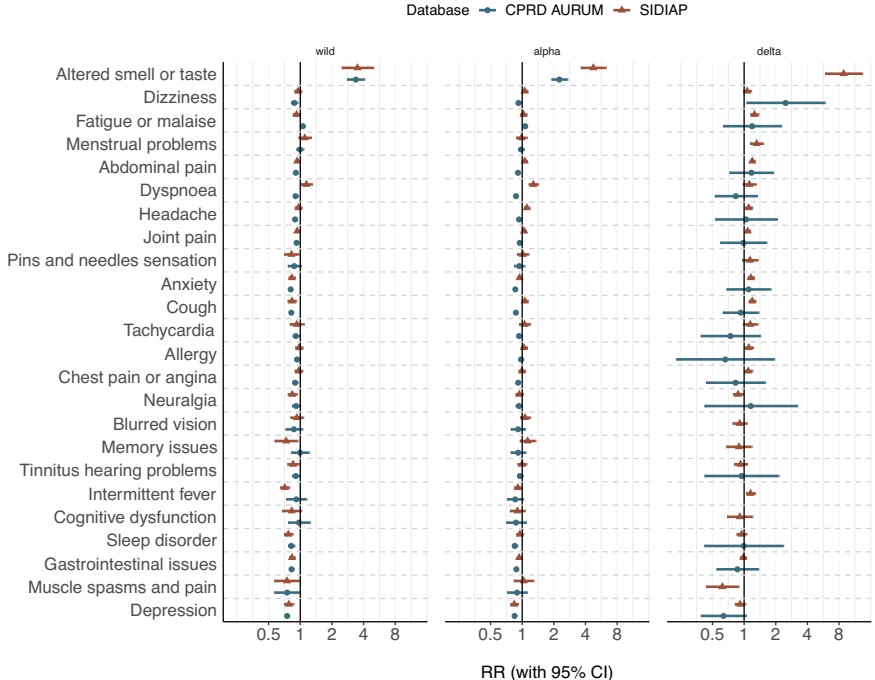

**Fig. 4 | Rate ratios (RRs) for pre-defined post-acute COVID-19 symptoms, comparing 1:3 matched SARS-CoV-2 infections to first SARS-CoV-2 negative tests, stratified for wave of predominant SARS-CoV-2 variant.** SIDIAP Sistema d'Informació per al Desenvolupament de la Investigació en Atenció Primària, CPRD Clinical Practice Research Datalink, RR Rate ratio, 95% CI 95% confidence intervals. RR and 95% CI were calculated among 229,086 and 332,276 SARS-CoV-2 infections and matched 591,145 and 912,745 first SARS-CoV-2 negative tests in SIDIAP and CPRD, respectively.

after SARS-CoV-2 infections than in controls in both Spain and the UK. Dyspnoea was substantially increased after SARS-CoV-2 infection relative to controls in Spanish data, and persisting fatigue/malaise was seen among UK participants following COVID-19.

We report a consistent increase in the risk of persistent symptoms after reinfection compared to first infection. All post-acute COVID-19 symptoms mentioned in the WHO clinical case definition appeared more common after reinfection than after a first infection, after matching by age, sex and date of infection.

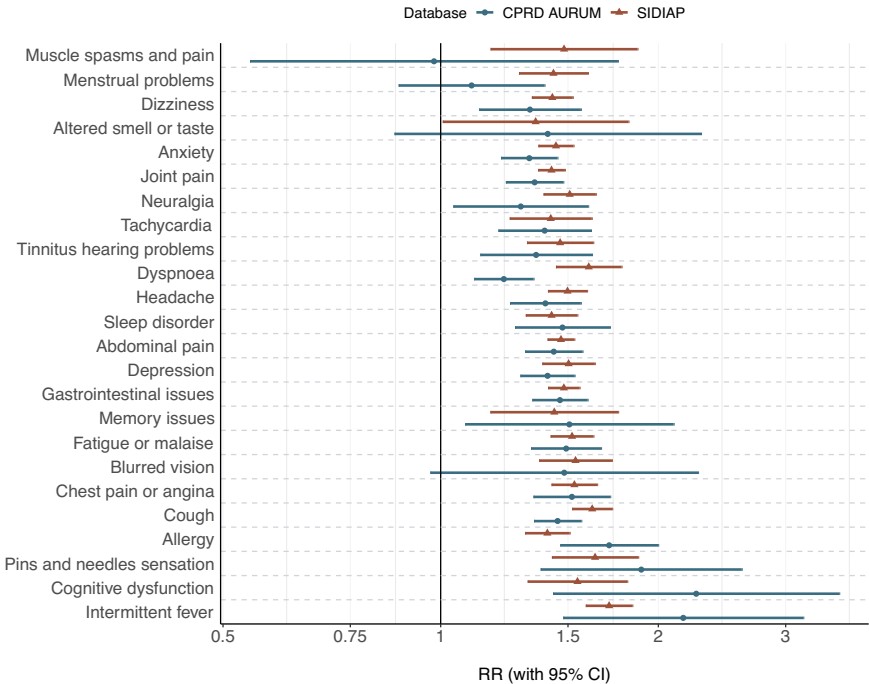

**Fig. 5 | Rate ratios (RRs) for pre-defined post-acute COVID-19 symptoms comparing 1:3 matched re-infections to first infections.** SIDIAP Sistema d'Informació per al Desenvolupament de la Investigació en Atenció Primària, CPRD Clinical Practice Research Datalink, RR Rate ratio, 95% CI 95% confidence intervals. RR and 95% CI were calculated among 155,400 and 48,574 first SARS-CoV-2 infections, and 55,297 and 16,916 matched re-infections in SIDIAP and CPRD, respectively.

## Research in context

Persistent symptoms were common after SARS-CoV-2 infection but were prevalent to almost the same extent as in the general population during the same period. In line with our results, previous studies found a substantial proportion of non-infected people with records of similar symptoms[14]. This finding highlights the challenge in identifying long COVID, which could result in misclassification and difficulties in diagnosing long COVID.

While the prevalence of post-acute COVID-19 symptoms remained high in both countries, our results showed a decline in the proportion of people developing persistent symptoms after COVID-19 over time. The REACT-2 study, a representative community survey among adults in England, found a similar trend, with the prevalence of people experiencing at least one symptom >12 weeks following COVID-19 declining from 37.7% between September 2020 and February 2021 to 21.6% in May 2021[15]. This trend may be attributable to the effect of vaccines[16], previous immunity (i.e., reinfection), and differences in the pre-dominant variant[17]. However, shorter follow-up time available later in the pandemic, potential differences in testing practice shifting from prioritising severely ill people to wider testing for screening as well as non-systematically collection of symptoms carrying the risk for "reporting exhaustion" and a decreased reporting of the same persisting symptoms, may also account for the observed reduction in prevalence of post-acute COVID-19 symptoms over time. Similar declining trends were reported for the US beyond our study period[18].

Our study adds to previous research focusing on long-term complications following SARS-CoV-2 infection and frequencies of persistent symptoms. Hundreds of different symptoms have been reported in relation to COVID-19, and the Centre for Disease Control and Prevention recently highlighted that not all those self-reported symptoms were unique to COVID-19 or to post-COVID conditions[19]. Our study therefore focussed on those 25 symptoms, which the WHO highlighted in their Delphi consensus as particularly characteristic.

Previous studies compared some of these symptoms in people with and without COVID-19. Subramanian et al. [14] determined symptoms associated with COVID-19 after 12 weeks by comparing people with confirmed COVID-19 to propensity-score-matched controls without recorded or suspected SARS-CoV-2 infection in CPRD. No negative test was required for the control group. Similar to our study, they found that symptoms with a strong association with SARS-COV-2 included anosmia, shortness of breath at rest, and fatigue. However, although we found that only some symptoms were associated with COVID infection, Subramanian et al. found that the risk for all recorded symptoms was significantly increased after infection. Similar to our study, a previous study from the US found not all post-acute sequelae to be differential when comparing SARS-CoV-2 PCR positive tests with PCR negative controls, with only risk for anosmia, cardiac dysrhythmias, diabetes, genitourinary conditions, malaise and fatigue and non-specific chest pain being significantly increased[20].

The effect of reinfection on the severity and persistence of COVID-19 symptoms remains a topic of great interest. Our multinational study assessed the effect of reinfection on the risk of post-acute COVID-19 symptoms as defined by the WHO. Our results showed an increased risk for post-acute COVID-19 symptoms following re-infection, suggesting that people with re-infections remain at risk for developing persisting symptoms. Previous studies on this topic are scarce and discordant: A previous study on post-acute complications and organ system disorders in people following first or reinfection in the US Veterans Health Administration database[21] reported a twofold increased risk for at least one sequela, which was consistent regardless of vaccination status. However, this previous study did not investigate long COVID as an outcome, and the study population was not representative of the general population. Another study by the Office of National Statistics based on data from the COVID-19 Infection Survey, however, reported a 28% lower risk for new-onset, self-reported post-acute COVID-19 symptoms among adults after a second COVID-19 infection compared with a first infection[22].

## Strengths and weaknesses

COVID-19 datasets are burdened with systemic limitations as the pandemic placed significant strain on the global healthcare system. As broad testing was not available in most countries in early 2020, we began our study period in September 2020, excluding the first wave of the pandemic. With widespread issues in testing capacities to meet public demand and the advent of self-administered tests, underreporting of infections is expected across all pandemic waves. Some reinfections may therefore have been misclassified as first infections. Likewise, we expect underreporting of clinical symptoms as people might not have been seen by a clinician, particularly for milder symptoms, during infection peaks and after re-infection if symptoms were similar as for previous infections. Our study period predominantly covers the earlier waves of the pandemic, and hence, symptom presentations following infections with later variants, including omicron or XBB, might vary.

Aside from differences in time of subject inclusion, differences in healthcare, with more virtual clinical work in the UK than in Spanish primary care practice, may explain the small difference observed in rates of post-acute COVID-19 symptoms between SIDIAP and CPRD. Long COVID is a new condition, with its definition developing over time. With clinical awareness still evolving, reporting bias in the recording practice of characteristic symptoms cannot be ruled out. A systematic and comprehensive collection and reporting of long COVID symptoms would be needed to overcome this limitation.

Our study also has strengths. We included two large population-based databases from different European regions with primary-care-based universal public healthcare. CPRD AURUM and SIDIAP provide high-quality data for research and are representative of their respective populations[23]. Prior research evaluated individual symptom prevalence associated with post COVID-19 conditions, but no other study has compared these estimates between different countries and care settings in a matched cohort, looking at the implications of grouping by first infection, re-infection, and negative test. Our methodology allowed us to ascertain general population averages during pandemic times and quantify overall health status after lockdowns or other public health policies, regardless of COVID-19 status. Despite the many challenges that long COVID patients report facing in gaining clinical recognition of their symptoms, an increasingly consistent clinical presentation is evident in this multi-database view for SARS-Cov-2 infections during the wild, alpha and delta waves.

Systematic reviews have shown that data harmonisation is fundamental to improving the clinical utility of findings[24]. A strength of our research is the use of a common data model (OMOP CDM) and shared conventions in data harmonisation, allowing for consistent representation of clinical information despite heterogeneous source systems.

Waves of post-acute COVID-19 symptoms were observable following community transmission during the first two years of the pandemic, affecting predominantly women and young adults. However, the proportion of COVID-19 cases affected by persistent symptoms declined more recently, which could be due to a mixture of growing immunity due to vaccines and natural immunity. Our findings showed an increased risk for developing persistent symptoms following SARS-COV-2 re-infections compared to first infections, suggesting that people remain at risk for developing persistent symptoms despite previously build-up immunity. We identified 'altered smell and taste' as a key symptom that can help to differentiate people living with post-COVID-19 conditions. More work is needed to improve the existing definition of long COVID to enhance future trials into the efficacy of vaccines and antivirals to prevent and/or manage this disease.

## Methods

### Data sources, study design and study period

We conducted a population-based descriptive cohort study using primary care electronic health records from England, UK and Catalonia, Spain.

Primary care electronic health records in England were obtained from the Clinical Practice Research Datalink (CPRD) AURUM, which comprises 20% of the population in the UK[25,26]. Spanish data were obtained from the Information System for Research in Primary Care (SIDIAP; www.sidiap.org) database, which captures more than 75% of the population living in Catalonia, a region in the northeast of Spain[23]. SIDIAP was linked to hospital discharge records from public and private hospitals in Catalonia (Conjunt Mínim Bàsic de Dades d'Alta Hospitalària, CMBD-AH)[27]. Both databases include information on demographics, clinical diagnoses, and laboratory tests, including SARS-CoV-2 reverse transcription polymerase chain reaction (RT-PCR) tests. SIDIAP also captures SARS-CoV-2 antigen tests performed at public healthcare facilities. Although information on SARS-CoV-2 antigen testing may appear in CPRD, the counts are expected to be incomplete.

The databases were standardised to the Observational Medical Outcomes Partnership Common Data Model (OMOP CDM)[28], allowing the same analytical code to be applied without sharing individual data.

The study period spanned from 1 September 2020 to the end of data availability, i.e. January 2022 for CPRD and March 2022 for SIDIAP, where data was censored to avoid misclassification due to changes in COVID testing policies.

### Study population

We defined three non-mutually-exclusive COVID-19 cohorts—(1) all COVID-19 cases, (2) first SARS-CoV-2 infections, and (3) SARS-CoV-2 reinfections—and two negative-test comparator cohorts—(1) first/earliest SARS-CoV-2 negative tests and (2) all SARS-CoV-2 negative tests.

COVID-19 cases were identified using positive SARS-CoV-2 antigen and RT-PCR tests, using the test date as the index date. COVID-19 was defined as infections without a record of SARS-CoV-2 infections in the previous 42 days. First infections were defined as SARS-CoV-2 infections without any prior history of COVID-19. Reinfections were defined as SARS-CoV-2 infections that were not identified as a first infection.

The two negative-test comparator cohorts were identified using negative SARS-CoV-2 antigen and RT-PCR tests, using the test date as the index date. Individuals included in these cohorts were required to have a SARS-CoV-2 negative test result without a clinical COVID-19 diagnosis or positive SARS-CoV-2 test result before the index date and up to 120 days after the index date. SARS-CoV-2 negative tests were defined as records of a negative test without a record of a prior negative test 42 days before the index date (similar to the definition used for COVID-19 cases). First, SARS-CoV-2 negative tests were defined as SARS-CoV-2 negative tests without any prior history of a negative test. Concept lists used to define the COVID-19, and test-negative cohorts are available from https://github.com/oxford-pharmacoepi/LongCOVIDWP1A.

All cohorts included individuals aged ≥18 years with ≥180 days of data visibility available before the index date. Individuals with an influenza clinical diagnosis or positive test result for influenza 42 days before or on the index date were excluded. To ensure sufficient follow-up to assess post-acute COVID-19 symptoms and reduce survival bias, we only included individuals with ≥120 days of follow-up, i.e. with an index date ≥120 days before the end of data availability. All cohorts were followed until the occurrence of the first event of interest, death, new SARS-CoV-2 infection, or a record of a COVID-19 clinical diagnosis, influenza infection (positive test result or clinical diagnosis), one year of follow-up, or end of data availability. In SIDIAP, cohorts were also censored on 28 March 2022, as national guidelines no longer recommended testing all suspected COVID-19 cases after that date.

### Post-acute COVID-19 symptoms

We identified post-acute COVID-19 symptoms included in the WHO clinical case definition of "post COVID-19 condition"[7] based on SNOMED codes in the OMOP CDM mapped respective datasets. Twenty-five symptoms were included: abdominal pain, allergy, altered

smell and/or taste, anxiety, blurred vision, chest pain, cognitive dysfunction, cough, depression, dizziness, dyspnoea, fatigue or malaise, gastrointestinal issues (acid reflux, constipation, or diarrhoea), headache, intermittent fever, joint pain, memory issues, menstrual problems, muscle spasms or pain, neuralgia, pins and needles sensation, post-exertional fatigue, sleep disorder, tachycardia, and tinnitus and hearing problems. Separate code lists were developed for each symptom and reviewed independently by three clinicians (https://github.com/oxford-pharmacoepi/LongCOVIDWP1A). Quality checks were conducted to systematically identify missing codes (Cohort-Diagnostics R package)[29].

The WHO definition was then adapted to identify post-acute COVID-19 symptoms in primary care data. Long COVID was defined as having at least one record of any of the pre-defined symptoms between 90 and 365 days after the date of SARS-CoV-2 infection and no record of that symptom 180 days before the index date. This 180-day washout window prior to the index date was included to reduce misclassification of pre-existing symptoms, e.g. anxiety, which was re-recorded after the index date. Figure S1 illustrates the algorithm.

For the negative-test cohorts, we anchored the algorithm at the date of the negative test to compare the proportion of people with symptoms. In sensitivity analyses, we also reported "ongoing symptomatic COVID-19"[30], defined as having at least one record of one of the symptoms ≥28 days after the index date.

### Statistical analyses
We developed a common analytical code, which was subsequently run locally in OMOP CDM mapped CPRD AURUM and SIDIAP, respectively. All results are reported separately by database. We described and compared baseline characteristics (age groups [≤34, 35–49, 50–64, 65–79 and ≥80], sex, calendar time [trimester, waves], COVID-19 vaccine status [unvaccinated and number of vaccine doses received], and co-morbidities) for people with SARS-CoV-2 infection and negative-test comparator cohorts. We compared the proportion of people with post-acute COVID-19 symptoms (≥90 and ≥28 days) across the five cohorts. We calculated monthly incidence rates per 100,000 person-years for COVID-19 and post-acute COVID-19 symptoms in the general population (i.e. all people in the database without a record of post-acute COVID-19 symptoms) and among people with COVID-19.

To understand which of the pre-specified symptoms would be more differential for long COVID, we matched people with SARS-CoV-2 infections and negative controls (first negative tests, and any negative test, respectively) by 5-year age group, sex (female, male), SARS-CoV-2 test (antigen or PCR), and index week (ratio 1:3). Rate ratios with 95% confidence intervals for each symptom are presented in forest plots. We similarly matched people with first and re-infections (ratio 3:1) and compared rate ratios for post-acute COVID-19 symptoms at ≥90 and ≥28 days.

Analyses were performed locally in compliance with all applicable data privacy laws. Analyses were conducted in R (version 4.3.1). All analytical code is available at https://github.com/oxford-pharmacoepi/LongCOVIDWP1A.

### Patient and public involvement
A patient and public representative were involved in planning the overarching project and helped contextualise the study results using their patient perspective.

### Ethics approval
The study was approved by the relevant Institution Review Boards: the CPRD's Research Data Governance Process (Protocol No. 21_000557), the Clinical Research Ethics Committee of Fundació Institut Universitari per a la recerca a l'Atenció Primària de Salut Jordi Gol i Gurina (IDIAPJGol) (approval number 4R22/133). No informed consent of individuals was required as the study only used secondary data.

### Reporting summary
Further information on research design is available in the Nature Portfolio Reporting Summary linked to this article.

## Data availability
CPRD data were obtained under the CPRD multi-study license held by the University of Oxford after Research Data Governance (RDG) approval. Direct data sharing is not allowed due to privacy laws. Access to CPRD can be requested from CPRD directly and is subject to protocol approval via CPRD's RDG process (https://cprd.com/data-access). Following current European and national law, SIDIAP data are only available for researchers participating in this study. However, researchers from public institutions can request data from SIDIAP if they comply with certain requirements. Further information is available online (https://www.sidiap.org/index.php/en/solicituds-en) or by contacting SIDIAP (sidiap@idiapjgol.org).

## Code availability
All analytical code is available at https://github.com/oxford-pharmacoepi/LongCOVIDWP1A.

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

## Acknowledgements

This study is based in part on data from the Clinical Practice Research Datalink (CPRD) obtained under license from the UK Medicines and Healthcare Products Regulatory Agency. We thank the patients who provided these data and the NHS who collected the data as part of their care and support. We also thank the Institut Català de la Salut (ICS) and the Programa d'analítica de dades per a la recerca i la innovació en salut (PADRIS) for providing access to the different data sources accessible through SIDIAP. All interpretations, conclusions and views expressed in this publication are those of the author(s) alone. We acknowledge English language editing by Dr Jennifer A de Beyer, Centre for Statistics in Medicine, University of Oxford. *Funding* This work is independent research funded by the National Institute for Health and Care Research (NIHR) (COV-LT2-0006). DPA's group received partial support from the Oxford NIHR Biomedical Research Centre. The views expressed in this publication are those of the author(s) and not necessarily those of NIHR or the Department of Health and Social Care. E.R. was supported by Instituto de Salud Carlos III (ISCIII; Río Hortega 2020, CM20/00174).

## Author contributions

D.P.A., M.C. and A.M.J. led the conceptualisation of the study, with contributions from E.R. and K.K. K.K., D.P.A. and A.M.J. led the phenotyping of long COVID symptoms. A.D. mapped and curated CPRD data. E.R., N.M.B. and M.C. conducted the statistical analyses on the respective databases. D.P.A., E.R., T.D.S., N.T., L.M., R.P. and A.M.J. clinically interpreted the results. K.K., E.R. and A.M.J. wrote the first draft of the paper. All authors read, contributed to, and approved the last version of the paper. D.P.A. and A.M.J. obtained the funding for this research.

## Competing interests

D.P.A.'s department has received grant/s from Amgen, Chiesi–Taylor, Lilly, Janssen, Novartis, and UCB Biopharma. His research group has received consultancy fees from Astra Zeneca and UCB Biopharma. Amgen, Astellas, Janssen, Synapse Management Partners and UCB Biopharma have funded or supported training programmes organised by DPA's department. LM reports grants from Grifols, speaker fees from AstraZeneca and Gilead and participation in Advisory Boards for Gilead. NT is supported by the Norwegian Research Council (grant No. 288696) and has received internationalisation support from UiO: Life Science and travel grants from the Norwegian Research Council. The remaining authors declare no competing interests.
