## [Peer Review File · Nature Communications]

The burden of post-acute COVID-19 symptoms in a multinational network cohort analysisREVIEWER COMMENTS

Reviewer #1 (Remarks to the Author):

-What are the noteworthy results?

The key findings for me are that risk for altered smell/taste, dyspnoea, and fatigue were consistently higher in long COVID patient vs controls, which may help differentiate people living with Long Covid, from those living with symptoms associated with the implementation of wider pandemic measures, or other non-COVID causes.

In addition, these data show a reduction in risk of ongoing symptoms following reinfection, which is in contrast to previous limited literature (e.g. Bove 2022 Nat Med). The authors do a good job of explaining why their data show a different trend.

-Will the work be of significance to the field and related fields? How does it compare to the established literature? If the work is not original, please provide relevant references.

The analysis of multinational cohorts with matched controls is novel. But it would be useful if the authors could articulate how their work differs from Subramanian et al (2022), who conducted similar analyses using CPRD.

-Does the work support the conclusions and claims, or is additional evidence needed?

Yes, although the following statement in the conclusion is a little speculative given the data from this study is unable to determine the relative contribution of natural immunity and vaccination on long covid following reinfection.

"Our findings suggest that natural immunity plays a key role as reinfections had a much lower risk of long COVID than first infections."

-Are there any flaws in the data analysis, interpretation and conclusions? Do these prohibit publication or require revision?

No major flaws. It would be useful if the authors could expand on potential reasons for the stark difference in direction of rate ratios between CPRD and SIDIAP, for pre-defined long COVID symptoms, comparing 1:3 matched COVID-19 infections to first SARS-CoV-2 negative tests.

There's a lot of data presented in this paper and I feel that the results could use a little narrowing of focus on the key messages - which are differentiating symptoms and reduction in risk of ongoing symptoms following reinfection. I would remove the results on incidence rates over time, as these are well characterised elsewhere and somewhat distract from the key messages.

-Is the methodology sound? Does the work meet the expected standards in your field?

Yes, but this should be checked by a statistician.

-Is there enough detail provided in the methods for the work to be reproduced?

Yes.

Reviewer #2 (Remarks to the Author):

Thank you for a generally well-written manuscript. While there are already a substantial amount of studies addressing the occurrence of long-COVID symptoms in the general population, there is still very little information available on the potential influence of re-infections, and thus this manuscript is a welcome addition to the existing literature.

However, there are some unclarities, which makes it a bit difficult to judge parts of the results and parts of the methodology (see details below)

Please, remember that COVID-19 is the name of a disease and not a pathogen, so there is strictly speaking no such thing as "COVID-19 infection". Either it is "COVID-19" or "SARS-CoV-2 infection". Additionally, infections cannot "record..." anything, so please use infected persons or similar instead, if this is what you want to say.

Specific comments/questions:

- Line 1-3: Max. number of words for titles in NC is 15 words
- Line 37: The manuscript does not describe "the effect of persisting COVID-19 symptoms", so please consider rewording? The effect would e.g. be reduced quality of life, sick leave or something else caused by the symptoms.
- Line 37-49: A general comment to the abstract – preferably major factors such as study design, study population and data source should be indicated, so if e.g. the number of participants, that it is mainly non-hospitalized COVID-19 patients and that the study is registry-based could be added, it would be great
- Line 83-84 (and the introduction in general): Maybe consider cutting down on the general stuff that has been common knowledge for some time now, and instead be a bit more specific and e.g. outline the results of some of those studies, who actually did compare clinical definitions across studies and countries, e.g. the Global Burden of Disease Long COVID Collaborators Estimated Global Proportions of Individuals With Persistent Fatigue, Cognitive, and Respiratory Symptom Clusters Following Symptomatic COVID-19 in 2020 and 2021 | Neurology | JAMA | JAMA Network (could also be done in the discussion instead)
- Line 85-93: Very long and a bit awkward paragraph to state that using a control group is necessary – please consider shortening
- Line 97-99/in general: The wording is a bit unclear here: If you by "the general population" mean the test-negative group, then please state that. Additionally, it seems a bit wrong to talk about long-COVID in this group, since these will naturally not fulfill the case-definition – consider calling it "symptoms associated with Long-COVID in the general non-infected population" or similar (provided this is what you mean).
- Line 116/Figure 1: It is not entirely clear, why specifically persons "With a negative test 42 days before" need to be excluded from the "COVID-19 negative tests, initial records" group? Shouldn't it either be exclusion of all, where it is not the first negative test, or none (i.e. only exclusion of those, who had ever obtained a positive test result)?
- Line 118/Table 1/in general: 1) Have you considered using periods of SARS-CoV-2 variant predominance instead of semesters? This might be more relevant in terms of interpretation of results, 2) Why first SARS-CoV-2 negative test, and not just any negative test from an individual not preceded by a positive test result? Were there test-requirements in place, that could made this distinction necessary? Dependent on the test requirements and policies in the countries in question, one could fear that this would introduce bias over time, since those most likely to be tested e.g. healthcare workers or persons with underlying diseases, who has to test to go to hospital would no more be eligible for inclusion as the study period proceeds? For context it could also be nice to know the test-incidences in the two countries, the development in these over time

as well as the test criteria (Do these also play a role in the observed differences between the two countries?).

- Line 141: "Rates of long COVID declined over time in both databases" – yes, and with vaccinations, I guess it is expected for the SIDIAP data, but how would you explain this development for the CPRD data, that seems to include only three months of testing during 2020? Do you believe, that this is an actual decline, or not e.g. caused by a change in test strategy/incidence (inclusion of more less ill persons etc)?

- Line 144/Figure 2: 1) Given the linguistic confusion in the first part of this manuscript, where it seems like "COVID-19 infections" have been used instead of "test-positives" in line 133 and "the general population" has been used synonymous with test-negatives in lines 97-9, it is unclear whether this figure literally shows what it says on the labels in the right side of the figure i.e. number of COVID-19 infections, incidence of long-COVID in the general population (which would normally be interpreted as among the infected, but non-hospitalised part) and incidence of long-COVID in relation to the no. of infections or whether it instead shows incidence of long-COVID symptoms among the infected, symptoms among those never infected, and symptoms among the infected adjusted for symptoms among the uninfected? Please clarify and add appropriate labels.

- Line 169. Consider replacing "with" with "for".

- Line 175: Table references need updating, S9 and S10 instead?

- Line 186: The baseline characteristics are not in table S10, but S8, so please update.

- Line 191/Figure 4: Why are only so few CPRD symptoms included? Too few observations?

- Line 200: The results of most studies within the field do indeed indicate that long-COVID mainly affects women and the young or middle-aged. However, when looking at your results in Fig. 2A and B, bottom panel row, I am not sure that this is the conclusion, I would draw? Here it looks like, that given one has been infected, the older age groups (and maybe also males) are more likely to be infected?

Given that the results are based on primary care records, and previous studies have indicated gender-differences in medical-seeking behavior, this is a little surprising. This makes me wonder, whether the male part of the study population could have been more severely affected during the acute-phase? If information about hospitalization as a proxy for severity of acute infection are available, it could be interesting to see the results stratified by this.

- Line 202-203: 1) Other studies have indicated that compared to previous variants the Omicron variant are less likely to cause alterations in smell and taste both in the acute and post-acute phase, so one could wonder whether this is still the most distinctive symptoms today, 2) In general – do you still think that it is realistic/useful to aim for identifying one set of distinctive symptoms? Since COVID-19 is a systematic disease, it is expected that post-acute symptoms might occur from multiple organ systems. Additionally, several non-exclusive biological mechanisms have been suggested, and others studies have started looking at different clusters/phenotypes of long-COVID symptoms, so this approach might seem a bit out of touch with the most recent knowledge within the area. If you don't want to go into it, then maybe better just to state that in the present study these were the most frequently observed symptoms? (Also relevant in relation to lines 276-278 as well as 289-291).

- Line 205-207: Please improve this sentence

- Line 209: To might best knowledge, the WHO Delphi consensus does not contain a definition of long-COVID symptoms as such, other than they defined some symptoms to be included in the expert elicitation, i.e. considered for inclusion the case definition. However in the end, the case definition was not restricted to these.

- Lines 241-250: Regarding the re-infected population – do you feel confident that those who had lingering symptoms first time, and get the same symptoms second time, would seek medical care again?
- Lines 405-6: Please, explain the need for two negative comparator cohorts. It is not clear to the reader, when the second one (all negative tests) is used, since in both results tables, the comparator group is listed as first negative test.
- Lines 408-9: Why is the criteria only within 120 days after the index date and not no positive test within the period of follow-up at all? With a median follow-up time of 358 days in the SIDIAP dataset (table 1), wouldn't there be a risk of positive tests not accounted for leading to the inclusion of acute as well as post-acute COVID-19 related symptoms?

Reviewer #3 (Remarks to the Author):

Overall: This is a well-written and clear manuscript on the important topic of Long COVID, including predominant symptoms, comparison to the COVID-negative concurrent populations, and emergence of Long COVID among COVID-19 re-infected individuals. It includes two large geographic populations—which supposedly makes it multi-national, but the authors never merge the populations, so that distinction of multi-national is hard to follow.

There are a few major criticisms that should be addressed:

1. The matching seems incomplete. The authors did not match by sex, race/ethnicity, where the diagnosis was made (outpatient might be different than inpatient) or the database itself, if they were to merge. These would have made the analyses stronger. At least, they need to acknowledge this in the Limitations section.
2. Greater comparisons with prior studies is needed, to place the results presented here in greater context. Nature Communications has published other work on this topic (see Horberg et al, Nature Communications, 2022 Oct 12;13(1):5822. doi: 10.1038/s41467-022-33573-6).
3. While might be minor, Supplementary Tables S2 and S3 should be in the main text.
4. Figures 3 and 4 would benefit by including a bar for any symptom. That would really give a sense of overall incidence. And again, the 2 cohorts (CPRD and SIDIAP) should be merged with more complete matching.
5. With such extensive databases, it would have been good to control for pre-existing conditions. Many of these conditions described may have been there pre-COVID. While reactivation of these symptoms may have been by COVID infection, they really are not incident events for Long COVID. It would have been good to have that accounted for. If the authors did do that, it's not clear from the text.

Minor points:

1. Abstract: Please include the matching criteria in the limited abstract.
2. Abstract: Should be made clear that you didn't merge the databases to do the analysis.
3. The last line of Introduction paragraph should have some references.
4. The US, for example, had more widely available PCR testing by May 2020. The results here nearly miss most of the first wave of infections.
5. The authors limited infection definition to 42 days (see line 402). Most would use 90 days or so. I have no comments on the supplementary tables or figures.

Point-by-point response to Reviewer comments

We thank the Editor and Reviewers 1, 2, and 3 for their time reviewing our manuscript and their very helpful comments and feedback.

We replied to all comments in detail below and revised the manuscript accordingly.

REVIEWER COMMENTS

Reviewer #1:

- *What are the noteworthy results?*

The key findings for me are that risk for altered smell/taste, dyspnoea, and fatigue were consistently higher in long COVID patient vs controls, which may help differentiate people living with Long Covid, from those living with symptoms associated with the implementation of wider pandemic measures, or other non-COVID causes. In addition, these data show a reduction in risk of ongoing symptoms following reinfection, which is in contrast to previous limited literature (e.g. Bowe 2022 Nat Med). The authors do a good job of explaining why their data show a different trend.

Thank you very much for the positive feedback.

- *Will the work be of significance to the field and related fields? How does it compare to the established literature? If the work is not original, please provide relevant references.*
The analysis of multinational cohorts with matched controls is novel. But it would be useful if the authors could articulate how their work differs from Subramanian et al (2022), who conducted similar analyses using CPRD.

Subramanian et al. used CPRD AURUM to identify symptoms associated with SARS-CoV-2 infection beyond 3 months in non-hospitalised adults. People with confirmed SARS-CoV-2 infection were propensity-score matched to adults “with no recorded evidence” of SARS-CoV-2.

Therefore, the main difference between Subramanian’s study and our work is the comparison group. While we chose people who explicitly tested negative and therefore had no COVID-19 at index date to be our reference group, the previous study did not ascertain that their comparison group did not have (suspected/tested/diagnosed) COVID-19 but relied on the absence of a positive test/diagnosis. This is of particular concern as the main results presented in their previous study include the first wave of the pandemic, where wider testing was not available and PCR tests were restricted to be used only the hospital setting. Therefore, a substantial number of people are expected to have been infected with SARS-CoV-2 between March and September 2020 without a recorded diagnosis/PCR test for COVID-19. Our study overcomes these limitations by choosing the tested-negative cohort for comparison and restricting the study period to Sept 2020 onwards when wider testing was available.

We added a sentence to the discussion to highlight this important difference:

p. 15 line 244: “Subramanian et al.²¹ determined symptoms associated with COVID-19 after 12 weeks by comparing people with confirmed COVID-19 to propensity-score-matched controls without recorded or suspected COVID-19 infection in CPRD. No negative test was required for the control group.”

In addition, we provide additional follow-up with CPRD AURUM data now, covering up to 01/2022 compared to 04/2021. This allows us to stratify for period of variant predominance, which has not been done before. Figure 4 presents results from stratified analyses.

Lastly, we compare first infections and re-infection with respect to persisting COVID-19 symptoms, which has not been reported on before in European data. Our findings from updated analyses with longer follow-up and among a study population with higher proportions

of vaccinated people show increased risk for persisting symptoms after re-infection compared to first infections. This finding is in line with previous literature from Bowe et al. We updated the discussion section accordingly.

- *Does the work support the conclusions and claims, or is additional evidence needed?*
Yes, although the following statement in the conclusion is a little speculative given the data from this study is unable to determine the relative contribution of natural immunity and vaccination on long covid following reinfection: "Our findings suggest that natural immunity plays a key role as reinfections had a much lower risk of long COVID than first infections."

We rephrased the sentence according to the updated results. P. 17 line 308: "Our findings showed an increased risk for developing persistent symptoms following SARS-COV-2 re-infections compared to first infections, suggesting that people remain at risk for developing persistent symptoms despite previously build-up immunity. "

- *Are there any flaws in the data analysis, interpretation and conclusions? Do these prohibit publication or require revision?* No major flaws. It would be useful if the authors could expand on potential reasons for the stark difference in direction of rate ratios between CPRD and SIDIAP, for pre-defined long COVID symptoms, comparing 1:3 matched COVID-19 infections to first SARS-CoV-2 negative tests.

Thank you for this comment.

As suggested by the Editor, we provide additional follow-up for CPRD AURUM data to allow for a better comparison with SIDIAP (Data availability: CPRD up until 01/22, SIDIAP up until 03/22). In addition, we added "sex" as a matching criterion as suggested by Reviewer 3.

In the new results, the differences in RR between SIDIAP and CPRD is substantially smaller, likely due to more comparable follow-up time and coverage of the pandemic waves for the 2 databases.

- There's a lot of data presented in this paper and I feel that the results could use a little narrowing of focus on the key messages - which are differentiating symptoms and reduction in risk of ongoing symptoms following reinfection. I would remove the results on incidence rates over time, as these are well characterised elsewhere and somewhat distract from the key messages.

Thank you for this comment. We discussed this suggestion in detail between the co-authors and – while we agree that the results sections comprises a lot of results – we think that population-level incidence estimates of long COVID provide important context for the subsequent results. We would therefore prefer to keep Fig 2 in the main manuscript, unless the Editors prefer us to remove it.

- *Is the methodology sound? Does the work meet the expected standards in your field?*
Yes, but this should be checked by a statistician.

Thank you.

- *Is there enough detail provided in the methods for the work to be reproduced?*
Yes.

Thank you.

Reviewer #2:

Thank you for a generally well-written manuscript. While there are already a substantial amount of studies addressing the occurrence of long-COVID symptoms in the general population, there is still very little information available on the potential influence of re-infections, and thus this manuscript is a welcome addition to the existing literature.

Thank you for this positive feedback.

However, there are some unclarities, which makes it a bit difficult to judge parts of the results and parts of the methodology (see details below):

- Please, remember that COVID-19 is the name of a disease and not a pathogen, so there is strictly speaking no such thing as “COVID-19 infection”. Either it is “COVID-19” or “SARS-CoV-2 infection”. Additionally, infections cannot “record...” anything, so please use infected persons or similar instead, if this is what you want to say.

We edited the manuscript accordingly to use “SARS-CoV-2 infection” and SARS-CoV-2 negative test consistently throughout the manuscript.

- Specific comments/questions:
 - Line 1-3: Max. number of words for titles in NC is 15 words

Thank you for that remark. We shortened the title to meet the word restriction. “The burden of post-acute COVID-19 symptoms: a multinational analysis including SARS-CoV-2 infections, reinfections, and test-negative controls.”

- Line 37: The manuscript does not describe “the effect of persisting COVID-19 symptoms”, so please consider rewording? The effect would e.g. be reduced quality of life, sick leave or something else caused by the symptoms.

We rephrased the respective sentence to read as follows: p. 2 line 36: “We conducted a cohort study to characterise post-acute COVID-19 symptoms and identified key symptoms associated with persistent disease”

- Line 37-49: A general comment to the abstract – preferably major factors such as study design, study population and data source should be indicated, so if e.g. the number of participants, that it is mainly non-hospitalized COVID-19 patients and that the study is registry-based could be added, it would be great

We edited to abstract to include the study design (line 36: “[...] we conducted a cohort study”) and provided additional details on the type of routinely-collected data used in this study (line 37: “Using primary care electronic health records from Spain and UK”).

- Line 83-84 (and the introduction in general): Maybe consider cutting down on the general stuff that has been common knowledge for some time now, and instead be a bit more specific and e.g. outline the results of some of those studies, who actually did compare clinical definitions across studies and countries, e.g. the Global Burden of Disease Long COVID Collaborators Estimated Global Proportions of Individuals With Persistent Fatigue, Cognitive, and Respiratory Symptom Clusters Following Symptomatic COVID-19 in 2020 and 2021 | Neurology | JAMA | JAMA Network (could also be done in the discussion instead)

We shortened the introduction and provide context on the frequency of long COVID by citing the reference suggested by the Reviewer.

- Line 85-93: Very long and a bit awkward paragraph to state that using a control group is necessary – please consider shortening

Thank you for this comment. We shortened the respective paragraph as suggested.

- Line 97-99/in general: The wording is a bit unclear here: If you by “the general population” mean the test-negative group, then please state that.

Thank you for that remark. We actually refer to the general population here, as we calculated incidence of long COVID using all people in the database without a previous record of long COVID as the denominator.

We edited the introduction and methods section to provide more clarity for this analysis:

P. 3 Line 89: “We first estimated age- and sex-specific incidence rates of persisting symptoms in the general population and among people with confirmed COVID-19 over time. Subsequently, we investigated which of the 25 symptoms the WHO mentions in their clinical

case definition are more specific to Long COVID by comparing the occurrence of each symptom among COVID-19 patients and people who tested negative in the same week”

P. 21 Line 491: “We calculated monthly incidence rates per 100,000 person-years for COVID-19 and post-acute COVID-19 symptoms in the general population (i.e. all people in the database without a record of post-acute COVID-19 symptoms) [...]”.

- Line 97-99/in general: Additionally, it seems a bit wrong to talk about long-COVID in this group, since these will naturally not fulfil the case-definition – consider calling it “symptoms associated with Long-COVID in the general non-infected population” or similar (provided this is what you mean).

Thank you for this comment. As this sentence refers to the general population (and not to the tested negative control group) the term long COVID is accurate in this context. We calculated monthly incidence rates of people with SARS-CoV-2 infection followed by persistent symptoms at >28 or >90days.

We separated the two sentences describing (1) the calculation of long COVID incidence rates and (2) persistent symptoms in people tested positive and negative for SARS-CoV-2 respectively, by adding the word “Subsequently...” to provide more clarity.

P. 3 Line 89: “We first estimated age- and sex-specific incidence rates of persisting symptoms in the general population and among people with confirmed COVID-19 over time. Subsequently, we investigated which of the 25 symptoms the WHO mentions in their clinical case definition are more specific to Long COVID by comparing the occurrence of each symptom among COVID-19 patients and people who tested negative in the same week”

- Line 116/Figure 1: It is not entirely clear, why specifically persons “With a negative test 42 days before” need to be excluded from the “COVID-19 negative tests, initial records” group? Shouldn’t it either be exclusion of all, where it is not the first negative test, or none (i.e. only exclusion of those, who had ever obtained a positive test result)?

As described in the methods section (p. 20 line 437) we defined COVID-19 “as infections without a record of SARS-CoV-2 infections in the previous 42 days.” This definition including a washout-period of 6 weeks was used to separate subsequent infections, as people could have multiple positive pcr/tests clustered around the same SARS-CoV-2 infection. For the test negative comparison we applied the same criteria as for the COVID-19 cohort, including the 42-days washout period.

We added an explanation for this criterion in the methods section:

P. 20 Line 445: “SARS-CoV-2 negative tests were defined as records of a negative test without a record of a prior negative test 42 days before the index date (similar to the definition used for COVID-19 cases).”

- Line 118/Table 1/in general:
 - 1) Have you considered using periods of SARS-CoV-2 variant predominance instead of semesters? This might be more relevant in terms of interpretation of results

Thank you for this suggestion. We report periods of SARS-CoV-2 variant predominance in Table 1 now. Moreover, we provide a new Figure 4 showing rate ratios for long COVID symptoms in matched SARS-CoV-2 infections to first SARS-CoV-2 negative tests stratified for periods of predominant SARS-CoV-2 variants.

- 2) Why first SARS-CoV-2 negative test, and not just any negative test from an individual not preceded by a positive test result? Were there test-requirements in place, that could made this distinction necessary? Dependent on the test requirements and policies in the countries in question, one could fear that this would introduce bias over time, since those most likely to be tested e.g. healthcare workers or persons with underlying diseases, who has to test to go to hospital would no more be eligible for inclusion as the study period proceeds? For context it could also be nice

to know the test-incidences in the two countries, the development in these over time as well as the test criteria (Do these also play a role in the observed differences between the two countries?).

We chose “first SARS-CoV-2 negative test” as the primary comparator to avoid overrepresentation of people with frequent testing, e.g. healthcare workers, we chose the first test to allow one person to only contribute one test. First negative tests were also selected as we matched them to first SARS-CoV-2 infection.

However, as highlighted by Reviewer 2, choosing first negative test only can introduce bias over time: As first negative tests were predominantly conducted at the start of the study period only a smaller number of first tests were available for matching later in the study/pandemic. We therefore also conducted matching to “any negative test” for sensitivity analyses.

We edited the following sentence in the results section to provide a clearer explanation of the sensitivity analysis:

P. 10 line 170: “Results from sensitivity analyses matching to any negative test instead of the first negative test are included in Supplementary Tables S8 and S9 and Figures S4 and S5 and showed similar findings.”

Both analyses are mentioned in the methods section:

P. 21 line 495: “We matched people with SARS-CoV-2 infections and negative controls (first negative tests, and any negative test, respectively)”.

- Line 141: “Rates of long COVID declined over time in both databases” – yes, and with vaccinations, I guess it is expected for the SIDIAP data, but how would you explain this development for the CPRD data, that seems to include only three months of testing during 2020? Do you believe, that this is an actual decline, or not e.g. caused by a change in test strategy/incidence (inclusion of more less ill persons etc)?

This is an important point. We re-run our analyses with a later datacut for CPRD AURUM, which provides us with longer follow-up up until 01/2022 instead of 04/2021. As shown in the new Figure 2, we see a similar trend in both SIDIAP and CPRD. We agree with the Reviewer that this is likely due to vaccination, and following the COVID-19 waves in the general population.

- Line 144/Figure 2: 1) Given the linguistic confusion in the first part of this manuscript, where it seems like “COVID-19 infections” have been used instead of “test-positives” in line 133 and “the general population” has been used synonymous with test-negatives in lines 97-9, it is unclear whether this figure literally shows what it says on the labels in the right side of the figure i.e. number of COVID-19 infections, incidence of long-COVID in the general population (which would normally be interpreted as among the infected, but non-hospitalised part) and incidence of long-COVID in relation to the no. of infections or whether it instead shows incidence of long-COVID symptoms among the infected, symptoms among those never infected, and symptoms among the infected adjusted for symptoms among the uninfected? Please clarify and add appropriate labels.

We separated the two sentences in line 92-94 by including the term “Subsequently” to avoid confusion and emphasise that we refer to two different analyses. Incidence rates of post-acute COVID-19 symptoms were calculated in the general population while the comparison of frequency persisting symptoms was conducted in people with SARS-CoV-2 infection and matched test negative controls.

We acknowledge that the cohorts for the general population and tested-negative people largely overlap as most people got tested for SARS-CoV-2 at some point during the pandemic. However, as we applied different inclusion criteria they are not exactly the same.

The labels in Figure 2 were therefore correct.

- Line 169. Consider replacing “with” with “for”.

Thanks, we edited the sentence accordingly.

- Line 175: Table references need updating, S9 and S10 instead?

Thanks! We updated the references to the tables in the supplement accordingly.

- Line 186: The baseline characteristics are not in table S10, but S8, so please update.

We updated the reference to the baseline characteristics table accordingly.

- Line 191/Figure 4: Why are only so few CPRD symptoms included? Too few observations?

Exactly. We initially included all symptoms for analysis, and in line with CPRDs data privacy protection regulations we only included those symptoms with frequencies ≥ 5 in the plot and reported this subset in tables S11 and S12.

However, since we updated the manuscript to include longer follow-up for CPRD AURUM, more symptoms could be displayed.

- Line 200: The results of most studies within the field do indeed indicate that long-COVID mainly affects women and the young or middle-aged. However, when looking at your results in Fig. 2A and B, bottom panel row, I am not sure that this is the conclusion, I would draw? Here it looks like, that given one has been infected, the older age groups (and maybe also males) are more likely to be infected? Given that the results are based on primary care records, and previous studies have indicated gender-differences in medical-seeking behavior, this is a little surprising. This makes me wonder, whether the male part of the study population could have been more severely affected during the acute-phase? If information about hospitalization as a proxy for severity of acute infection are available, it could be interesting to see the results stratified by this.

Thank you for this comment. This sentence refers to post-acute COVID-19 symptoms in the general population (panel 2 row), which in both databases shows higher incidence rates in females relative to males, and in younger age groups. However, we agree with the Reviewer’s comment in that among people who had COVID-19, higher proportions of persistent symptoms were observed in older age groups.

We agree that differences in healthcare-seeking behaviour – and, related to that, the probability of persisting symptoms being recorded – might differ between age- and gender. Unfortunately, we don’t have information on the severity of the infection (i.e. hospitalisation).

- Line 202-203: 1) Other studies have indicated that compared to previous variants the Omicron variant are less likely to cause alterations in smell and taste both in the acute and post-acute phase, so one could wonder whether this is still the most distinctive symptoms today.

Thank you for this comment. We stratified our analyses for time of “predominant variant”, covering the wild type, alpha and delta variant. Results are shown in Fig 4 in the revised manuscript. Altered smell/taste was the most distinctive symptom for all 3 variants.

As we required a minimum follow-up of 120-days after infection to allow for the assessments of persistent symptoms, only few infections recorded during the omicron wave were included in this study. We could therefore not conduct stratified analyses focussing on the omicron wave.

- 2) In general – do you still think that it is realistic/useful to aim for identifying one set of distinctive symptoms? Since COVID-19 is a systematic disease, it is expected that post-acute symptoms might occur from multiple organ systems. Additionally, several non-exclusive biological mechanisms have been suggested, and other studies have started looking at different clusters/phenotypes of long-COVID symptoms, so this approach might seem a bit out of touch with the most recent knowledge within the area.

Thank you for raising this important point. We very much agree with the Reviewer in that more than 200 different symptoms affecting multiple organ systems have been reported in the context of post-acute COVID-19 complications and symptoms. However, previous research suggests that many of those are related, and that having a core set of symptoms can be very very useful to facilitate future research, and improve the characterisation and understanding of long COVID.

The WHO published a list of 25 symptoms that – at the time – were among the most frequently reported symptoms and those which put a major burden to the affected patients. However, as many of those symptoms are prevalent among the overall population and more and more literature on long COVID is becoming available suggesting that the presentation of both acute COVID-19 and persistent symptoms differ with respect of predominant variants, we strongly believe that understanding which symptoms are particularly distinctive is very important to inform future research.

- If you don't want to go into it, then maybe better just to state that in the present study these were the most frequently observed symptoms? (Also relevant in relation to lines 276-278 as well as 289-291)

Thank you for this suggestion. We discussed this in detail among the co-authors and agreed that the preferred to stick to the current wording as we believe this would most accurately reflect our analyses. We matched first SARS-CoV-2 first infections to first tested negative infections. Subsequently, we calculated rate ratios as the number of people with the respective persisting symptom after first infection among all people with a first infection divided by the frequency of that symptom in tested-negative controls among all matched people with a first negative test. The aim was to understand which of the WHO-listed symptoms were more common after SARS-CoV-2 infection relative to tested-negative controls. Therefore, we do rank symptoms based on frequency.

- Line 205-207: Please improve this sentence:

We edited this section accordingly following the updated results from the new analyses incl. matching for sex and the new CPRD AURUM dataset.

- Line 209: To might best knowledge, the WHO Delphi consensus does not contain a definition of long-COVID symptoms as such, other than they defined some symptoms to be included in the expert elicitation, i.e considered for inclusion the case definition. However in the end, the case definition was not restricted to these.

We agree with the Reviewers comment that the WHO clinical case definition lists “fatigue, shortness of breath, cognitive dysfunction and others ” as common symptoms in their definition, and specify “others” in a “full list of described symptoms included in the (delphi consensus) surveys”.*

We adapted the wording in the introduction and discussion section as follows:

P. 14 line 216: “All post-acute COVID-19 symptoms mentioned in the WHO clinical case definition”

P. 3 line 94: “We investigated which of the 25 symptoms the WHO mentions in their clinical case definition are particularly specific to Long COVID”

- Lines 241-250: Regarding the re-infected population – do you feel confident that those who had lingering symptoms first time, and get the same symptoms second time, would seek medical care again?

This is a very important point. We expect underreporting of symptoms, for both first and re-infections, as people might not have been seen by clinicians during the infection peaks, and people might have hesitated to attend their GPs for milder symptoms or recurring symptoms that have resolved previously.

We expanded the limitation section accordingly:

P. 16 line 267: Likewise, we expect underreporting of clinical symptoms as people might not have been seen by a clinician, particularly for milder symptoms, during infection peaks and after re-infection if symptoms were similar as for previous infections.”

- Lines 405-6: Please, explain the need for two negative comparator cohorts. It is not clear to the reader, when the second one (all negative tests) is used, since in both results tables, the comparator group is listed as first negative test.

We chose “first SARS-CoV-2 negative test” as the primary comparator to avoid overrepresentation of people with frequent testing, e.g. healthcare workers. However, choosing first negative test only can introduce bias over time: As first negative tests were predominantly conducted at the start of the study period only a smaller number of first tests were available for matching later in the study. We therefore conducted a sensitivity analysis matching first SARS-CoV-2 infections to “any negative test”.

We edited the respective sentence in the results section as follows:

P. 10 line 170: “Results from sensitivity analyses matching to any negative test instead of the first negative test are included in Supplementary Tables S8 and S9 and Figures S4 and S5 and showed similar findings.”

- Lines 408-9: Why is the criteria only within 120 days after the index date and not no positive test within the period of follow-up at all?

Thank you for this question. We required a minimum 120 days of follow-up to reduce survival bias when assessing persisting symptoms at ≥ 90 days. However, as most people at some point during the pandemic tested positive for SARS-CoV-2 we would have excluded a substantial number of people had we required “no positive test within the whole follow-up period”. To retain sample size while reducing survival bias, we opted for the 120 days definition and subsequently censored follow-up for tested negative controls at the time the respective person had their first positive SARS-Cov-2 test or COVID-19 diagnosis.

We added a rationale for the 120 days definition in the methods section:

P. 20 line 453: “To ensure sufficient follow-up to assess post-acute COVID-19 symptoms and reduce survival bias, we only included individuals with ≥ 120 days of follow-up.”

- With a median follow-up time of 358 days in the SIDIAP dataset (table 1), wouldn't there be a risk of positive tests not accounted for leading to the inclusion of acute as well as post-acute COVID-19 related symptoms?

We agree that acute COVID-19 symptoms could be misclassified as ongoing symptoms from previous SARS-CoV-2 infections. We therefore censored follow-up for people with a first infection at the time of a new, subsequent SARS-CoV-2 infection (Methods section p. 20 line 446) to avoid misclassification of symptoms.

Reviewer #3:

Overall: This is a well-written and clear manuscript on the important topic of Long COVID, including predominant symptoms, comparison to the COVID-negative concurrent populations, and emergence of Long COVID among COVID-19 re-infected individuals. It includes two large geographic populations — which supposedly makes it multi-national, but the authors never merge the populations, so that distinction of multi-national is hard to follow.

There are a few major criticisms that should be addressed:

- 1. The matching seems incomplete. The authors did not match by sex, race/ethnicity, where the diagnosis was made (outpatient might be different than inpatient) or the database itself, if they were to merge. These would have made the analyses stronger. At least, they need to acknowledge this in the Limitations section.

Thank you for raising this important point.

Conducting this study in 2 different databases from 2 European countries allowed us to compare findings across different healthcare settings, which we consider a particular strength of the study. Therefore, we did not aim to pool (merge) results or data, but conducted separate analyses running the same analytical code locally without sharing patient-level data.

P. 21 line 485, “all results were reported separately by database”

Following the Reviewer’s recommendation we re-run our analyses adding sex as a criterion for matching.

We, however, decided not to use ethnicity for matching as there is a high degree of missingness of ethnicity recordings in both SIDIAP and CPRD. While we did not match on test setting (outpatient, in hospital) we indeed included the type of test conducted, namely pcr and lft, for matching. During the study period testing was widely available and not only restricted to the hospital setting anymore, and lfd-tests were widely used for home testing.

2. Greater comparisons with prior studies is needed, to place the results presented here in greater context. Nature Communications has published other work on this topic (see Horberg et al, Nature Communications, 2022 Oct 12;13(1):5822. doi: 10.1038/s41467-022-33573-6).

We expanded the discussion section to put our findings in greater context and add the citation of Hoberg et al as suggested by the Reviewer.:

P. 15 line 239: “Hundreds of different symptoms have been reported in relation to COVID-19, and the Centre of Disease Control and Prevention recently highlighted that not all those self-reported symptoms were unique to COVID-19 or to post-COVID conditions”

P. 15 line 251: “Similar to our study, a previous study from the US [Hoberg et al.] found not all post-acute sequelae to be differential when comparing SARS-CoV-2 PCR positive tests with PCR negative controls, with only risk for anosmia, cardiac dysrhythmias, diabetes, genitourinary conditions, malaise and fatigue and non-specific chest pain being significantly increased”.

P. 15 line 256: “The effect of reinfection on the severity and persistence of COVID-19 symptoms remains a topic of great interest, and our multinational study is the first to assess the effect of reinfection on the risk of post-acute COVID-19 symptoms as defined by the WHO. Our results showed an increased risk for post-acute COVID-19 symptoms following reinfection, suggesting that people with re-infections remain at risk for developing persisting symptoms. Previous studies on this topic are scarce and discordant: A previous study on post-acute complications and organ system disorders in people following first or reinfection in the US Veterans Health Administration database reported an 2-fold increased risk for at least one sequela, which was consistent regardless of vaccination status. However, this previous study did not investigate long COVID as an outcome, and the study population was not representative for the general population. Another study by the Office of National Statistics based on data from the COVID-19 Infection Survey, however, reported a 28% lower risk for new-onset, self-reported post-acute COVID-19 symptoms among adults after a second COVID-19 infection compared with a first infection after”.

- 3. While might be minor, Supplementary Tables S2 and S3 should be in the main text.

Thank you for this suggestion. Supplementary Tables S2 and S3 contain the distributions of symptoms across the SARS-CoV-2 infected and tested negative cohorts before matching.

While we think it is important to have these tables included in the supplement for descriptive purposes, we prefer to only report the results from matched analyses in the main manuscript. Table 1 and Table S1 highlight differences in key demographics (age, sex, type of SARS-CoV-2 test done) and the comparison of frequencies of persistent symptoms in these unmatched cohorts required great caution considering bias and confounding in unbalanced cohorts.

- 4. Figures 3 and 4 would benefit by including a bar for any symptom. That would really give a sense of overall incidence. And again, the 2 cohorts (CPRD and SIDIAP) should be merged with more complete matching.

Thank you for this suggestion. In this analysis, we aimed to understand which of the WHO-listed symptoms were particularly differential for long COVID. We therefore provide Rate Ratios only for the individual symptoms, but not overall. However, if the Editor strongly feels adding a bar for the overall rate ratio of “any” post-acute COVID-19 symptom would be preferable we can run an additional analysis in the datasets and provide this information.

- 5. With such extensive databases, it would have been good to control for pre-existing conditions. Many of these conditions described may have been there pre-COVID. While reactivation of these symptoms may have been by COVID infection, they really are not incident events for Long COVID. It would have been good to have that accounted for. If the authors did do that, it’s not clear from the text.

We defined long COVID as having at least one record of any of the pre-defined symptoms between 90 and 365 days after the date of SARS-CoV-2 infection and no record of that symptom 180 days before the index date. (Methods section line 474).

We added the following sentence to provide more clarity:

Line 476: “This 180-days washout window prior to index date was included to reduce misclassification of pre-existing symptoms, e.g. anxiety, which were re-recorded after index date.”

- Minor points: 1. Abstract: Please include the matching criteria in the limited abstract.

Thank you for this suggestion. As the word count of the abstract is limited we could unfortunately not add the matching criteria to the abstract. However, we have now included the matching criteria in the results section to provide them earlier in the manuscript: line 152: “We matched 1:3 by age group, sex, type of test (antigen or PCR) and index week [...].”

- 2. Abstract: Should be made clear that you didn’t merge the databases to do the analysis.

Unfortunately, it was not possible to add this information to the abstract due to the restricted word count.

We edited the respective sentence in the Statistical Analyses paragraph of the “Methods” section to highlight that analyses were carried out separately for CPRD and SIDIAP:

P. 21 Line 484: “We developed a common analytical code, which was subsequently run locally in OMOP CDM mapped CPRD AURUM and SIDIAP, respectively. All results are reported separately by database.”

We present results from CPRD and SIDIAP separately throughout the manuscript.

- 3. The last line of Introduction paragraph should have some references.

The last sentence of the introduction summarises an analysis conducted as part of the present study: we compared the occurrence of persistent symptoms after a first infection or after reinfection. This is a comparison within our study not to the literature. We rephrased the respective paragraph to make that clearer.

- 4. The US, for example, had more widely available PCR testing by May 2020. The results here nearly miss most of the first wave of infections.

Missing the first wave is intentionally as restricted testing capacities meant that not all people with suspected COVID-19 were tested. We therefore restricted to the time period when wider testing was available. In the UK and Spain wider testing became available in Summer 2020, with testing capacities substantially increasing from July 2020 in England.

This is explained in the discussion section line 272f. "As broad testing was not available in most countries in early 2020, we began our study period in September 2020, excluding the first wave of the pandemic."

- 5. The authors limited infection definition to 42 days (see line 402). Most would use 90 days or so.

We chose a 6 weeks' time window, in which we considered multiple recorded positive tests referring to the same SARS-CoV-2 infection. The 6 weeks window was based on recommendations from clinical experts.

- I have no comments on the supplementary tables or figures.

Thank you.

REVIEWERS' COMMENTS

Reviewer #1 (Remarks to the Author):

Thank you for addressing my previous remarks, I have no further comments.

Reviewer #2 (Remarks to the Author):

Thank you for these revisions.

I only have some minor comments here + some in the reviewer 2 section of the attached response to reviewers file.

In general:

*Please be consistent re. long COVID / Long COVID / post COVID-19 condition

*Line 43:

The wording seems a bit odd to be, consider rewording – maybe .. “the proportion of COVID-19 cases, where acute symptoms were followed by..” or “the proportion of COVID-19 cases affected by persistent ...” (if the word count does not allow the first)

*Lines 86-88 + 256-7: I think the “OCHESTRA” study (Clinical phenotypes and quality of life to define post-COVID-19 syndrome: a cluster analysis of the multinational, prospective ORCHESTRA cohort - eClinicalMedicine (thelancet.com)) and possibly also others might already have done this, i.e. the global burden of illness study I referred to first time

I still think the most important part of your study is the inclusion of reinfections, since the rest have been seen before and has become less novel now with the Omicron subvariants dominating

*Line 104: Consider replacing "provided" with "illustrated"? Alternatively, "an overview of the process is provided" or something like that

*Line 107: Seems a bit too generalizing just to write one year. Would prefer to see the exact number of days + range or quantiles

*Consider skipping table 1 and replacing it with table S1, since the data for the reinfected also ought to be in the MS and there is a lot of redundant information in these two tables

*Lines 164-65: Maybe write if differences between variants were observed or not - else there is not really any point in including these periods.

*Lines 232-37: Based on the results of your sensitivity analysis (and other studies) I would assume shorter follow-up should have lead to an increase, not a decrease?

*Lines 276-9: Yes, I think so too – the reported rates in the present study are rather low compared to what have been found in questionnaire or app based studies (inclusion of actual comparisons could be nice)

*Lines 296-7: "Despite the many challenges that long COVID patients report facing in gaining clinical recognition of their symptoms, an increasingly consistent clinical presentation is evident in this multi-database view." I think you somehow need to adress in the text, that your results might not be consistent with what is observed since the emergence of Omicron. In my opinion it is one of the biggest weaknesses of the study and not something that can be ignored.

Lines 312-4: Seems a bit odd, when you are not addressing the effect of vaccination on long-COVID anywhere else in the text. Maybe find a different closure remark?

Line 350: Please check ref. 12

Reviewer #3 (Remarks to the Author):

The authors have adequately addressed this reviewer's previous concerns. There are no additional concerns.

Point-by-point response to Reviewer comments

We thank the Editor and Reviewers 1, 2, and 3 for their time reviewing our manuscript and their very helpful comments and feedback.

We replied to all comments in detail below and revised the manuscript accordingly.

REVIEWER COMMENTS

Reviewer #1:

- *What are the noteworthy results?*

The key findings for me are that risk for altered smell/taste, dyspnoea, and fatigue were consistently higher in long COVID patient vs controls, which may help differentiate people living with Long Covid, from those living with symptoms associated with the implementation of wider pandemic measures, or other non-COVID causes. In addition, these data show a reduction in risk of ongoing symptoms following reinfection, which is in contrast to previous limited literature (e.g. Bowe 2022 Nat Med). The authors do a good job of explaining why their data show a different trend.

Thank you very much for the positive feedback.

- *Will the work be of significance to the field and related fields? How does it compare to the established literature? If the work is not original, please provide relevant references.*

The analysis of multinational cohorts with matched controls is novel. But it would be useful if the authors could articulate how their work differs from Subramanian et al (2022), who conducted similar analyses using CPRD.

Subramanian et al. used CPRD AURUM to identify symptoms associated with SARS-CoV-2 infection beyond 3 months in non-hospitalised adults. People with confirmed SARS-CoV-2 infection were propensity-score matched to adults “with no recorded evidence” of SARS-CoV-2.

Therefore, the main difference between Subramanian’s study and our work is the comparison group. While we chose people who explicitly tested negative and therefore had no COVID-19 at index date to be our reference group, the previous study did not ascertain that their comparison group did not have (suspected/tested/diagnosed) COVID-19 but relied on the absence of a positive test/diagnosis. This is of particular concern as the main results presented in their previous study include the first wave of the pandemic, where wider testing was not available and PCR tests were restricted to be used only the hospital setting. Therefore, a substantial number of people are expected to have been infected with SARS-CoV-2 between March and September 2020 without a recorded diagnosis/PCR test for COVID-19. Our study overcomes these limitations by choosing the tested-negative cohort for comparison and restricting the study period to Sept 2020 onwards when wider testing was available.

We added a sentence to the discussion to highlight this important difference:

p. 15 line 244: “Subramanian et al.²¹ determined symptoms associated with COVID-19 after 12 weeks by comparing people with confirmed COVID-19 to propensity-score-matched controls without recorded or suspected COVID-19 infection in CPRD. No negative test was required for the control group.”

In addition, we provide additional follow-up with CPRD AURUM data now, covering up to 01/2022 compared to 04/2021. This allows us to stratify for period of variant predominance, which has not been done before. Figure 4 presents results from stratified analyses.

Lastly, we compare first infections and re-infection with respect to persisting COVID-19 symptoms, which has not been reported on before in European data. Our findings from updated analyses with longer follow-up and among a study population with higher proportions

of vaccinated people show increased risk for persisting symptoms after re-infection compared to first infections. This finding is in line with previous literature from Bowe et al. We updated the discussion section accordingly.

- *Does the work support the conclusions and claims, or is additional evidence needed?*
Yes, although the following statement in the conclusion is a little speculative given the data from this study is unable to determine the relative contribution of natural immunity and vaccination on long covid following reinfection: "Our findings suggest that natural immunity plays a key role as reinfections had a much lower risk of long COVID than first infections."

We rephrased the sentence according to the updated results. P. 17 line 308: "Our findings showed an increased risk for developing persistent symptoms following SARS-COV-2 re-infections compared to first infections, suggesting that people remain at risk for developing persistent symptoms despite previously build-up immunity. "

- *Are there any flaws in the data analysis, interpretation and conclusions? Do these prohibit publication or require revision?* No major flaws. It would be useful if the authors could expand on potential reasons for the stark difference in direction of rate ratios between CPRD and SIDIAP, for pre-defined long COVID symptoms, comparing 1:3 matched COVID-19 infections to first SARS-CoV-2 negative tests.

Thank you for this comment.

As suggested by the Editor, we provide additional follow-up for CPRD AURUM data to allow for a better comparison with SIDIAP (Data availability: CPRD up until 01/22, SIDIAP up until 03/22). In addition, we added "sex" as a matching criterion as suggested by Reviewer 3.

In the new results, the differences in RR between SIDIAP and CPRD is substantially smaller, likely due to more comparable follow-up time and coverage of the pandemic waves for the 2 databases.

- There's a lot of data presented in this paper and I feel that the results could use a little narrowing of focus on the key messages - which are differentiating symptoms and reduction in risk of ongoing symptoms following reinfection. I would remove the results on incidence rates over time, as these are well characterised elsewhere and somewhat distract from the key messages.

Thank you for this comment. We discussed this suggestion in detail between the co-authors and – while we agree that the results sections comprises a lot of results – we think that population-level incidence estimates of long COVID provide important context for the subsequent results. We would therefore prefer to keep Fig 2 in the main manuscript, unless the Editors prefer us to remove it.

- *Is the methodology sound? Does the work meet the expected standards in your field?*
Yes, but this should be checked by a statistician.

Thank you.

- *Is there enough detail provided in the methods for the work to be reproduced?*
Yes.

Thank you.

Reviewer #2:

Thank you for a generally well-written manuscript. While there are already a substantial amount of studies addressing the occurrence of long-COVID symptoms in the general population, there is still very little information available on the potential influence of re-infections, and thus this manuscript is a welcome addition to the existing literature.

Thank you for this positive feedback.

However, there are some unclarities, which makes it a bit difficult to judge parts of the results and parts of the methodology (see details below):

- Please, remember that COVID-19 is the name of a disease and not a pathogen, so there is strictly speaking no such thing as “COVID-19 infection”. Either it is “COVID-19” or “SARS-CoV-2 infection”. Additionally, infections cannot “record...” anything, so please use infected persons or similar instead, if this is what you want to say.

We edited the manuscript accordingly to use “SARS-CoV-2 infection” and SARS-CoV-2 negative test consistently throughout the manuscript. OK

- Specific comments/questions:
 - Line 1-3: Max. number of words for titles in NC is 15 words

Thank you for that remark. We shortened the title to meet the word restriction. “The burden of post-acute COVID-19 symptoms: a multinational analysis including SARS-CoV-2 infections, reinfections, and test-negative controls.” OK

- Line 37: The manuscript does not describe “the effect of persisting COVID-19 symptoms”, so please consider rewording? The effect would e.g. be reduced quality of life, sick leave or something else caused by the symptoms.

We rephrased the respective sentence to read as follows: p. 2 line 36: “We conducted a cohort study to characterise post-acute COVID-19 symptoms and identified key symptoms associated with persistent disease” OK

- Line 37-49: A general comment to the abstract – preferably major factors such as study design, study population and data source should be indicated, so if e.g. the number of participants, that it is mainly non-hospitalized COVID-19 patients and that the study is registry-based could be added, it would be great

We edited to abstract to include the study design (line 36: “[...] we conducted a cohort study”) and provided additional details on the type of routinely-collected data used in this study (line 37: “Using primary care electronic health records from Spain and UK”). OK

- Line 83-84 (and the introduction in general): Maybe consider cutting down on the general stuff that has been common knowledge for some time now, and instead be a bit more specific and e.g. outline the results of some of those studies, who actually did compare clinical definitions across studies and countries, e.g. the Global Burden of Disease Long COVID Collaborators Estimated Global Proportions of Individuals With Persistent Fatigue, Cognitive, and Respiratory Symptom Clusters Following Symptomatic COVID-19 in 2020 and 2021 | Neurology | JAMA | JAMA Network (could also be done in the discussion instead).

We shortened the introduction and provide context on the frequency of long COVID by citing the reference suggested by the Reviewer. However, in several places in the manuscript you still claim that this is the first multinational comparison across cohorts, which I don't think is entirely correct? In addition to the study mentioned above there is also e.g. the ORCHESTRA cohort.

- Line 85-93: Very long and a bit awkward paragraph to state that using a control group is necessary – please consider shortening

Thank you for this comment. We shortened the respective paragraph as suggested. OK

- Line 97-99/in general: The wording is a bit unclear here: If you by “the general population” mean the test-negative group, then please state that.

Thank you for that remark. We actually refer to the general population here, as we calculated incidence of long COVID using all people in the database without a previous record of long COVID as the denominator. Here as well as elsewhere in the manuscript there still seem to some confusion re. terminology: To my best knowledge it is simply not valid to call symptoms among persons, who has not had COVID-19 for long COVID given that having had COVID-

19 is the most important part of the case definition and all symptoms are very general/unspecific. I would really encourage the use of, “symptoms associated with long-COVID” if not only taking about the infected population, where the use of the term “long COVID” is of course fine.

We edited the introduction and methods section to provide more clarity for this analysis:

P. 3 Line 89: “We first estimated age- and sex-specific incidence rates of persisting symptoms in the general population and among people with confirmed COVID-19 over time. Subsequently, we investigated which of the 25 symptoms the WHO mentions in their clinical case definition are more specific to Long COVID by comparing the occurrence of each symptom among COVID-19 patients and people who tested negative in the same week”

P. 21 Line 491: “We calculated monthly incidence rates per 100,000 person-years for COVID-19 and post-acute COVID-19 symptoms in the general population (i.e. all people in the database without a record of post-acute COVID-19 symptoms) [...]”.

- Line 97-99/in general: Additionally, it seems a bit wrong to talk about long-COVID in this group, since these will naturally not fulfil the case-definition – consider calling it “symptoms associated with Long-COVID in the general non-infected population” or similar (provided this is what you mean).

Thank you for this comment. As this sentence refers to the general population (and not to the tested negative control group) the term long COVID is accurate in this context. **Again – I have to disagree - it is not accurate, since if used correctly this term cannot be used for persons who haven't had COVID-19 (it would have been perfectly fine if you were only looking at the infected population and not the general population).** We calculated monthly incidence rates of people with SARS-CoV-2 infection followed by persistent symptoms at >28 or >90days.

We separated the two sentences describing (1) the calculation of long COVID incidence rates and (2) persistent symptoms in people tested positive and negative for SARS-CoV-2 respectively, by adding the word “Subsequently...” to provide more clarity.

P. 3 Line 89: “We first estimated age- and sex-specific incidence rates of persisting symptoms in the general population and among people with confirmed COVID-19 over time. Subsequently, we investigated which of the 25 symptoms the WHO mentions in their clinical case definition are more specific to Long COVID by comparing the occurrence of each symptom among COVID-19 patients and people who tested negative in the same week”

- Line 116/Figure 1: It is not entirely clear, why specifically persons “With a negative test 42 days before” need to be excluded from the “COVID-19 negative tests, initial records” group? Shouldn't it either be exclusion of all, where it is not the first negative test, or none (i.e. only exclusion of those, who had ever obtained a positive test result)?

As described in the methods section (p. 20 line 437) we defined COVID-19 “as infections without a record of SARS-CoV-2 infections in the previous 42 days.” This definition including a washout-period of 6 weeks was used to separate subsequent infections, as people could have multiple positive pcr/tests clustered around the same SARS-CoV-2 infection. For the test negative comparison we applied the same criteria as for the COVID-19 cohort, including the 42-days washout period.

We added an explanation for this criterion in the methods section:

P. 20 Line 445: “SARS-CoV-2 negative tests were defined as records of a negative test without a record of a prior negative test 42 days before the index date (similar to the definition used for COVID-19 cases).” **OK**

- Line 118/Table 1/in general:
 - 1) Have you considered using periods of SARS-CoV-2 variant predominance instead of semesters? This might be more relevant in terms of interpretation of results

Thank you for this suggestion. We report periods of SARS-CoV-2 variant predominance in Table 1 now. Moreover, we provide a new Figure 4 showing rate ratios for long COVID symptoms in matched SARS-CoV-2 infections to first SARS-CoV-2 negative tests stratified for periods of predominant SARS-CoV-2 variants. Fine that you included it, however, you don't really seem to be using it for anything? Did you observe any differences between the symptoms caused by the different variants?

- 2) Why first SARS-CoV-2 negative test, and not just any negative test from an individual not preceded by a positive test result? Were there test-requirements in place, that could have made this distinction necessary? Dependent on the test requirements and policies in the countries in question, one could fear that this would introduce bias over time, since those most likely to be tested e.g. healthcare workers or persons with underlying diseases, who have to go to hospital would no longer be eligible for inclusion as the study period proceeds? For context it could also be nice to know the test-incidences in the two countries, the development in these over time as well as the test criteria (Do these also play a role in the observed differences between the two countries?).

We chose "first SARS-CoV-2 negative test" as the primary comparator to avoid overrepresentation of people with frequent testing, e.g. healthcare workers, we chose the first test to allow one person to only contribute one test. First negative tests were also selected as we matched them to first SARS-CoV-2 infection. For this to make sense, shouldn't it then only be persons, who tested positive in their first test?

However, as highlighted by Reviewer 2, choosing first negative test only can introduce bias over time: As first negative tests were predominantly conducted at the start of the study period only a smaller number of first tests were available for matching later in the study/pandemic. We therefore also conducted matching to "any negative test" for sensitivity analyses.

We edited the following sentence in the results section to provide a clearer explanation of the sensitivity analysis:

P. 10 line 170: "Results from sensitivity analyses matching to any negative test instead of the first negative test are included in Supplementary Tables S8 and S9 and Figures S4 and S5 and showed similar findings."

Both analyses are mentioned in the methods section:

P. 21 line 495: "We matched people with SARS-CoV-2 infections and negative controls (first negative tests, and any negative test, respectively)."

- Line 141: "Rates of long COVID declined over time in both databases" – yes, and with vaccinations, I guess it is expected for the SIDIAP data, but how would you explain this development for the CPRD data, that seems to include only three months of testing during 2020? Do you believe, that this is an actual decline, or not e.g. caused by a change in test strategy/incidence (inclusion of more less ill persons etc)?

This is an important point. We re-run our analyses with a later dataset for CPRD AURUM, which provides us with longer follow-up up until 01/2022 instead of 04/2021. As shown in the new Figure 2, we see a similar trend in both SIDIAP and CPRD. We agree with the Reviewer that this is likely due to vaccination, and following the COVID-19 waves in the general population. Good

- Line 144/Figure 2: 1) Given the linguistic confusion in the first part of this manuscript, where it seems like "COVID-19 infections" have been used instead of "test-positives" in line 133 and "the general population" has been used synonymously with test-

negatives in lines 97-9, it is unclear whether this figure literally shows what it says on the labels in the right side of the figure i.e. number of COVID-19 infections, incidence of long-COVID in the general population (which would normally be interpreted as among the infected, but non-hospitalised part) and incidence of long-COVID in relation to the no. of infections or whether it instead shows incidence of long-COVID symptoms among the infected, symptoms among those never infected, and symptoms among the infected adjusted for symptoms among the uninfected? Please clarify and add appropriate labels.

We separated the two sentences in line 92-94 by including the term “Subsequently” to avoid confusion and emphasise that we refer to two different analyses. Incidence rates of post-acute COVID-19 symptoms were calculated in the general population while the comparison of frequency persisting symptoms was conducted in people with SARS-CoV-2 infection and matched test negative controls.

We acknowledge that the cohorts for the general population and tested-negative people largely overlap as most people got tested for SARS-CoV-2 at some point during the pandemic. However, as we applied different inclusion criteria they are not exactly the same.

The labels in Figure 2 were therefore correct. OK – I think part of the confusion re. fig.2 is caused by the wording in the heading, maybe consider reorganizing it a bit, e.g. to: “Incidence rates of COVID-19 in the general population and post-acute COVID-19 symptoms ≥ 90 days after infection among infected individuals by test date, stratified by sex and age group.”

- Line 169. Consider replacing “with” with “for”.

Thanks, we edited the sentence accordingly. OK

- Line 175: Table references need updating, S9 and S10 instead?

Thanks! We updated the references to the tables in the supplement accordingly. OK

- Line 186: The baseline characteristics are not in table S10, but S8, so please update.

We updated the reference to the baseline characteristics table accordingly. OK

- Line 191/Figure 4: Why are only so few CPRD symptoms included? Too few observations?

Exactly. We initially included all symptoms for analysis, and in line with CPRDs data privacy protection regulations we only included those symptoms with frequencies ≥ 5 in the plot and reported this subset in tables S11 and S12.

However, since we updated the manuscript to include longer follow-up for CPRD AURUM, more symptoms could be displayed. OK

- Line 200: The results of most studies within the field do indeed indicate that long-COVID mainly affects women and the young or middle-aged. However, when looking at your results in Fig. 2A and B, bottom panel row, I am not sure that this is the conclusion, I would draw? Here it looks like, that given one has been infected, the older age groups (and maybe also males) are more likely to be infected? Given that the results are based on primary care records, and previous studies have indicated gender-differences in medical-seeking behavior, this is a little surprising. This makes me wonder, whether the male part of the study population could have been more severely affected during the acute-phase? If information about hospitalization as a proxy for severity of acute infection are available, it could be interesting to see the results stratified by this.

Thank you for this comment. This sentence refers to post-acute COVID-19 symptoms in the general population (panel 2 row), which in both databases shows higher incidence rates in females relative to males, and in younger age groups. However, we agree with the Reviewer's comment in that among people who had COVID-19, higher proportions of persistent symptoms were observed in older age groups.

We agree that differences in healthcare-seeking behaviour – and, related to that, the probability of persisting symptoms being recorded – might differ between age- and gender. Unfortunately, we don't have information on the severity of the infection (i.e. hospitalisation). OK – my previous remark was caused by the somewhat awkward wording in fig 2 and the misunderstanding following from that

- Line 202-203: 1) Other studies have indicated that compared to previous variants the Omicron variant are less likely to cause alterations in smell and taste both in the acute and post-acute phase, so one could wonder whether this is still the most distinctive symptoms today.

Thank you for this comment. We stratified our analyses for time of “predominant variant”, covering the wild type, alpha and delta variant. Results are shown in Fig 4 in the revised manuscript. Altered smell/taste was the most distinctive symptom for all 3 variants.

As we required a minimum follow-up of 120-days after infection to allow for the assessments of persistent symptoms, only few infections recorded during the omicron wave were included in this study. We could therefore not conduct stratified analyses focussing on the omicron wave. OK, this is absolutely fair, but please also consider this in your conclusion – i.e. your results might not be that useful in terms of getting closer to a case definition, since it is fairly well-known that e.g. altered smell/taste are less dominant with Omicron (also seem to be the case for a couple of other symptoms).

- 2) In general – do you still think that it is realistic/useful to aim for identifying one set of distinctive symptoms? Since COVID-19 is a systematic disease, it is expected that post-acute symptoms might occur from multiple organ systems. Additionally, several non-exclusive biological mechanisms have been suggested, and other studies have started looking at different clusters/phenotypes of long-COVID symptoms, so this approach might seem a bit out of touch with the most recent knowledge within the area.

Thank you for raising this important point. We very much agree with the Reviewer in that more than 200 different symptoms affecting multiple organ systems have been reported in the context of post-acute COVID-19 complications and symptoms. However, previous research suggests that many of those are related, and that having a core set of symptoms can be very very useful to facilitate future research, and improve the characterisation and understanding of long COVID.

The WHO published a list of 25 symptoms that – at the time – were among the most frequently reported symptoms and those which put a major burden to the affected patients. However, as many of those symptoms are prevalent among the overall population and more and more literature on long COVID is becoming available suggesting that the presentation of both acute COVID-19 and persistent symptoms differ with respect of predominant variants, we strongly believe that understanding which symptoms are particularly distinctive is very important to inform future research.

- If you don't want to go into it, then maybe better just to state that in the present study these were the most frequently observed symptoms? (Also relevant in relation to lines 276-278 as well as 289-291)

Thank you for this suggestion. We discussed this in detail among the co-authors and agreed that the preferred to stick to the current wording as we believe this would most accurately reflect our analyses. We matched first SARS-CoV-2 first infections to first tested negative infections. Subsequently, we calculated rate ratios as the number of people with the respective

persisting symptom after first infection among all people with a first infection divided by the frequency of that symptom in tested-negative controls among all matched people with a first negative test. The aim was to understand which of the WHO-listed symptoms were more common after SARS-CoV-2 infection relative to tested-negative controls. Therefore, we do rank symptoms based on frequency. OK

- Line 205-207: Please improve this sentence:

We edited this section accordingly following the updated results from the new analyses incl. matching for sex and the new CPRD AURUM datacut. OK

- Line 209: To might best knowledge, the WHO Delphi consensus does not contain a definition of long-COVID symptoms as such, other than they defined some symptoms to be included in the expert elicitation, i.e considered for inclusion the case definition. However in the end, the case definition was not restricted to these.

We agree with the Reviewers comment that the WHO clinical case definition lists “fatigue, shortness of breath, cognitive dysfunction and others ” as common symptoms in their definition, and specify “others” in a “full list of described symptoms included in the (delphi consensus) surveys”.*

We adapted the wording in the introduction and discussion section as follows:

P. 14 line 216: “All post-acute COVID-19 symptoms mentioned in the WHO clinical case definition”

P. 3 line 94: “We investigated which of the 25 symptoms the WHO mentions in their clinical case definition are particularly specific to Long COVID” OK

- Lines 241-250: Regarding the re-infected population – do you feel confident that those who had lingering symptoms first time, and get the same symptoms second time, would seek medical care again?

This is a very important point. We expect underreporting of symptoms, for both first and re-infections, as people might not have been seen by clinicians during the infection peaks, and people might have hesitated to attend their GPs for milder symptoms or recurring symptoms that have resolved previously.

We expanded the limitation section accordingly:

P. 16 line 267: Likewise, we expect underreporting of clinical symptoms as people might not have been seen by a clinician, particularly for milder symptoms, during infection peaks and after re-infection if symptoms were similar as for previous infections.” OK

- Lines 405-6: Please, explain the need for two negative comparator cohorts. It is not clear to the reader, when the second one (all negative tests) is used, since in both results tables, the comparator group is listed as first negative test.

We chose “first SARS-CoV-2 negative test” as the primary comparator to avoid overrepresentation of people with frequent testing, e.g. healthcare workers. However, choosing first negative test only can introduce bias over time: As first negative tests were predominantly conducted at the start of the study period only a smaller number of first tests were available for matching later in the study. We therefore conducted a sensitivity analysis matching first SARS-CoV-2 infections to “any negative test”.

We edited the respective sentence in the results section as follows:

P. 10 line 170: “Results from sensitivity analyses matching to any negative test instead of the first negative test are included in Supplementary Tables S8 and S9 and Figures S4 and S5 and showed similar findings.” OK

- Lines 408-9: Why is the criteria only within 120 days after the index date and not no positive test within the period of follow-up at all?

Thank you for this question. We required a minimum 120 days of follow-up to reduce survival bias when assessing persisting symptoms at ≥ 90 days. However, as most people at some point during the pandemic tested positive for SARS-CoV-2 we would have excluded a substantial number of people had we required “no positive test within the whole follow-up period”. To retain sample size while reducing survival bias, we opted for the 120 days definition and subsequently censored follow-up for tested negative controls at the time the respective person had their first positive SARS-CoV-2 test or COVID-19 diagnosis.

We added a rationale for the 120 days definition in the methods section:

P. 20 line 453: “To ensure sufficient follow-up to assess post-acute COVID-19 symptoms and reduce survival bias, we only included individuals with ≥ 120 days of follow-up.” OK

- *With a median follow-up time of 358 days in the SIDIAP dataset (table 1), wouldn't there be a risk of positive tests not accounted for leading to the inclusion of acute as well as post-acute COVID-19 related symptoms?*

We agree that acute COVID-19 symptoms could be misclassified as ongoing symptoms from previous SARS-CoV-2 infections. We therefore censored follow-up for people with a first infection at the time of a new, subsequent SARS-CoV-2 infection (Methods section p. 20 line 446) to avoid misclassification of symptoms. OK

Reviewer #3:

Overall: This is a well-written and clear manuscript on the important topic of Long COVID, including predominant symptoms, comparison to the COVID-negative concurrent populations, and emergence of Long COVID among COVID-19 re-infected individuals. It includes two large geographic populations — which supposedly makes it multi-national, but the authors never merge the populations, so that distinction of multi-national is hard to follow.

There are a few major criticisms that should be addressed:

- 1. The matching seems incomplete. The authors did not match by sex, race/ethnicity, where the diagnosis was made (outpatient might be different than inpatient) or the database itself, if they were to merge. These would have made the analyses stronger. At least, they need to acknowledge this in the Limitations section.

Thank you for raising this important point.

Conducting this study in 2 different databases from 2 European countries allowed us to compare findings across different healthcare settings, which we consider a particular strength of the study. Therefore, we did not aim to pool (merge) results or data, but conducted separate analyses running the same analytical code locally without sharing patient-level data.

P. 21 line 485, “all results were reported separately by database”

Following the Reviewer's recommendation we re-run our analyses adding sex as a criterion for matching.

We, however, decided not to use ethnicity for matching as there is a high degree of missingness of ethnicity recordings in both SIDIAP and CPRD. While we did not match on test setting (outpatient, in hospital) we indeed included the type of test conducted, namely pcr and lft, for matching. During the study period testing was widely available and not only restricted to the hospital setting anymore, and lfd-tests were widely used for home testing.

2. Greater comparisons with prior studies is needed, to place the results presented here in greater context. Nature Communications has published other work on this topic (see Horberg et al, Nature Communications, 2022 Oct 12;13(1):5822. doi: 10.1038/s41467-022-33573-6).

We expanded the discussion section to put our findings in greater context and add the citation of Hoberg et al as suggested by the Reviewer.:

P. 15 line 239: "Hundreds of different symptoms have been reported in relation to COVID-19, and the Centre of Disease Control and Prevention recently highlighted that not all those self-reported symptoms were unique to COVID-19 or to post-COVID conditions"

P. 15 line 251: "Similar to our study, a previous study from the US [Hoberg et al.] found not all post-acute sequelae to be differential when comparing SARS-CoV-2 PCR positive tests with PCR negative controls, with only risk for anosmia, cardiac dysrhythmias, diabetes, genitourinary conditions, malaise and fatigue and non-specific chest pain being significantly increased".

P. 15 line 256: "The effect of reinfection on the severity and persistence of COVID-19 symptoms remains a topic of great interest, and our multinational study is the first to assess the effect of reinfection on the risk of post-acute COVID-19 symptoms as defined by the WHO. Our results showed an increased risk for post-acute COVID-19 symptoms following reinfection, suggesting that people with re-infections remain at risk for developing persisting symptoms. Previous studies on this topic are scarce and discordant: A previous study on post-acute complications and organ system disorders in people following first or reinfection in the US Veterans Health Administration database reported an 2-fold increased risk for at least one sequela, which was consistent regardless of vaccination status. However, this previous study did not investigate long COVID as an outcome, and the study population was not representative for the general population. Another study by the Office of National Statistics based on data from the COVID-19 Infection Survey, however, reported a 28% lower risk for new-onset, self-reported post-acute COVID-19 symptoms among adults after a second COVID-19 infection compared with a first infection after".

- 3. While might be minor, Supplementary Tables S2 and S3 should be in the main text.

Thank you for this suggestion. Supplementary Tables S2 and S3 contain the distributions of symptoms across the SARS-CoV-2 infected and tested negative cohorts before matching.

While we think it is important to have these tables included in the supplement for descriptive purposes, we prefer to only report the results from matched analyses in the main manuscript. Table 1 and Table S1 highlight differences in key demographics (age, sex, type of SARS-CoV-2 test done) and the comparison of frequencies of persistent symptoms in these unmatched cohorts required great caution considering bias and confounding in unbalanced cohorts.

- 4. Figures 3 and 4 would benefit by including a bar for any symptom. That would really give a sense of overall incidence. And again, the 2 cohorts (CPRD and SIDIAP) should be merged with more complete matching.

Thank you for this suggestion. In this analysis, we aimed to understand which of the WHO-listed symptoms were particularly differential for long COVID. We therefore provide Rate Ratios only for the individual symptoms, but not overall. However, if the Editor strongly feels adding a bar for the overall rate ratio of "any" post-acute COVID-19 symptom would be preferable we can run an additional analysis in the datasets and provide this information.

- 5. With such extensive databases, it would have been good to control for pre-existing conditions. Many of these conditions described may have been there pre-COVID. While reactivation of these symptoms may have been by COVID infection, they really are not incident events for Long COVID. It would have been good to have that accounted for. If the authors did do that, it's not clear from the text.

We defined long COVID as having at least one record of any of the pre-defined symptoms between 90 and 365 days after the date of SARS-CoV-2 infection and no record of that symptom 180 days before the index date. (Methods section line 474).

We added the following sentence to provide more clarity:

Line 476: “This 180-days washout window prior to index date was included to reduce misclassification of pre-existing symptoms, e.g. anxiety, which were re-recorded after index date.”

- Minor points: 1. Abstract: Please include the matching criteria in the limited abstract.

Thank you for this suggestion. As the word count of the abstract is limited we could unfortunately not add the matching criteria to the abstract. However, we have now included the matching criteria in the results section to provide them earlier in the manuscript: line 152: “We matched 1:3 by age group, sex, type of test (antigen or PCR) and index week [...].”

- 2. Abstract: Should be made clear that you didn't merge the databases to do the analysis.

Unfortunately, it was not possible to add this information to the abstract due to the restricted word count.

We edited the respective sentence in the Statistical Analyses paragraph of the “Methods” section to highlight that analyses were carried out separately for CPRD and SIDIAP:

P. 21 Line 484: “We developed a common analytical code, which was subsequently run locally in OMOP CDM mapped CPRD AURUM and SIDIAP, respectively. All results are reported separately by database.”

We present results from CPRD and SIDIAP separately throughout the manuscript.

- 3. The last line of Introduction paragraph should have some references.

The last sentence of the introduction summarises an analysis conducted as part of the present study: we compared the occurrence of persistent symptoms after a first infection or after reinfection. This is a comparison within our study not to the literature. We rephrased the respective paragraph to make that clearer.

- 4. The US, for example, had more widely available PCR testing by May 2020. The results here nearly miss most of the first wave of infections.

Missing the first wave is intentionally as restricted testing capacities meant that not all people with suspected COVID-19 were tested. We therefore restricted to the time period when wider testing was available. In the UK and Spain wider testing became available in Summer 2020, with testing capacities substantially increasing from July 2020 in England.

This is explained in the discussion section line 272f. “As broad testing was not available in most countries in early 2020, we began our study period in September 2020, excluding the first wave of the pandemic.”

- 5. The authors limited infection definition to 42 days (see line 402). Most would use 90 days or so.

We chose a 6 weeks' time window, in which we considered multiple recorded positive tests referring to the same SARS-CoV-2 infection. The 6 weeks window was based on recommendations from clinical experts.

- I have no comments on the supplementary tables or figures.

Thank you.

Point-by-point Response

We thank the Reviewers for their helpful feedback on our manuscript.

We adapted the manuscript according to the Reviewer's comments and response to them in detail below.

Page numbers and line numbers refer to the clean version of the revised manuscript.

REVIEWERS' COMMENTS

Reviewer #1

Thank you for addressing my previous remarks, I have no further comments.

Thank you.

Reviewer #2

Thank you for these revisions.

I only have some minor comments here + some in the reviewer 2 section of the attached response to reviewer's file.

In general:

*Please be consistent re. long COVID / Long COVID / post COVID-19 condition

Thank you. We aligned the wording throughout the manuscript as suggested.

*Line 43:

The wording seems a bit odd to be, consider rewording – maybe .. “the proportion of COVID-19 cases, where acute symptoms were followed by..” or “the proportion of COVID-19 cases affected by persistent ...” (if the word count does not allow the first)

Thank you for this suggestion. We adapted the sentence to read “the proportion of COVID-19 cases affected by persistent ...” as suggested.

*Lines 86-88 + 256-7: I think the “OCHESTRA” study (Clinical phenotypes and quality of life to define post-COVID-19 syndrome: a cluster analysis of the multinational, prospective ORCHESTRA cohort - eClinicalMedicine (thelancet.com)) and possibly also others might already have done this, i.e. the global burden of illness study I referred to first time I still think the most important part of your study is the inclusion of reinfections, since the rest have been seen before and has become less novel now with the Omicron subvariants dominating

Thank you. We rephrased the sentence to line 86 “*However, only few attempted to compare clinical definitions across multiple countries and/or sources of real-world data*”, and cite Gentilotti et al as suggested by the reviewer.

*Line 104: Consider replacing "provided" with "illustrated"? Alternatively, "an overview of the process is provided" or something like that

We replaced “*provided*” with “*illustrated*” as suggested.

*Line 107: Seems a bit too generalizing just to write one year. Would prefer to see the exact number of days + range or quantiles

We added the median duration in days. Interquartile ranges are presented in Table 1.

line 106: *Overall, follow-up was similar in SIDIAP and CPRD, with a median of 1 year (median 342 days and 364 days for SARS-CoV-2 infection cohorts in SIDIAP and CPRD, respectively, and 365 days for first negative test controls for both databases).*”

*Consider skipping table 1 and replacing it with table S1, since the data for the reinfected also ought to be in the MS and there is a lot of redundant information in these two tables

Thank you for this suggestion. We discussed this internally and prefer to present baseline characteristics for *all* SARS-CoV-2 infections for the main manuscript. This cohort was used for calculations of incidence rates, and thus provide important context for the manuscript. Table S1 presents descriptive statistics for first infections and re-infections *before* matching. However, as Figures 3 and 4 refer to the matched cohort, we believe it might be confusing to the reader to present baseline characteristics prior to matching. Baseline characteristics for matched cohorts are presented in the Supplement.

*Lines 164-65: Maybe write if differences between variants were observed or not - else there is not really any point in including these periods.

We state in line 152 that *“Results from stratification for wave of predominant variant are in line with the overall findings”*.

*Lines 232-37: Based on the results of your sensitivity analysis (and other studies) I would assume shorter follow-up should have led to an increase, not a decrease?

Shorter follow-up time available means that there is less time for the assessment of symptoms. People with persistent symptoms might wait a while until they present to their GP, and hence a shorter follow-up-time might impact the identification of cases.

*Lines 276-9: Yes, I think so too – the reported rates in the present study are rather low compared to what have been found in questionnaire or app based studies (inclusion of actual comparisons could be nice)

Thank you for this comment. We agree with the Reviewer that underreporting of both SARS-CoV-2 infections *and* persistent symptoms are a limitation in our study. For contextualisation, we report in line 199 on the REACT-2 study, a representative community survey among adults in England that showed the prevalence of people with persisting symptoms following COVID-19 declining from 37.7% between September 2020 and February 2021 to 21.6% in May 2021.

*Lines 296-7: "Despite the many challenges that long COVID patients report facing in gaining clinical recognition of their symptoms, an increasingly consistent clinical presentation is evident in this multi-database view." I think you somehow need to address in the text, that your results might not be consistent with what is observed since the emergence of Omicron. In my opinion it is one of the biggest weaknesses of the study and not something that can be ignored.

This is an important point. We expanded the limitation section to acknowledge this limitation: line 251: *“Our study period predominantly covers the earlier waves of the pandemic, and hence, symptom presentations following infections with later variants including omicron or XBB might vary.”*

line 269: *“Despite the many challenges that long COVID patients report facing in gaining clinical recognition of their symptoms, an increasingly consistent clinical presentation is evident in this multi-database view for SARS-Cov-2 infections during the wild, alpha and delta waves.”*

Lines 312-4: Seems a bit odd, when you are not addressing the effect of vaccination on long-COVID anywhere else in the text. Maybe find a different closure remark?

Thank you for this comment. We discuss potential reasons for a decline in the proportion of people developing persistent symptoms after COVID-19 over time, including but not limited to *“the effect of vaccines,¹⁶ previous immunity (i.e., reinfection), and differences in the predominant variant”* (line 202).

Line 350: Please check ref. 12

We fixed the reference. Thank you!

Reviewer #3

The authors have adequately addressed this reviewer's previous concerns. There are no additional concerns.

Thank you.